# EXPLORE WITH DYNAMIC MAP: GRAPH STRUCTURED REINFORCEMENT LEARNING

## ABSTRACT

In reinforcement learning, a map with states and transitions built using historical trajectories is often helpful in exploration and exploitation. Even so, learning and planning on such a map within a sparse environment remains a challenge. As a step towards this goal, we propose Graph Structured Reinforcement Learning (GSRL), which leverages graph structure in historical trajectories to slowly adjust exploration directions and rapidly update value function estimation with related experiences. GSRL constructs a dynamic graph on top of state transitions in the replay buffer based on historical trajectories and develops an attention strategy on the map to select an appropriate goal direction, which decomposes the task of reaching a distant goal state into a sequence of easier tasks. We also leverage graph structure to sample related trajectories for efficient value learning. Results demonstrate that GSRL can outperform the state-of-the-art algorithms in terms of sample efficiency on benchmarks with sparse reward functions.

## 1 INTRODUCTION

How can humans learn to solve tasks with complex structure and delayed and sparse feedback? Take a complicated navigation task as an example in which the goal is to go from a start state to an end state. A straightforward approach that one often takes is by decomposing this complicated task into a sequence of easier ones, identifying/forming some intermediate goals that help to get to the final goal, and finally choosing a route with the highest return among a poll of candidate routes that lead to the final goal. Despite recent success and progress in reinforcement learning (RL) approaches (Mnih et al., 2013; Fakoor et al., 2020; Hansen et al., 2018; Huang et al., 2019; Silver et al., 2016), there is still a huge gap between how humans and RL agents learn.

In many real-world applications, an RL agent only has access to sparse and delayed rewards, which by itself leads to two major challenges. **(C1)** The first one is how can an agent effectively learn from sparse and delayed rewards? One possible solution is to build a new reward function, known as an intrinsic reward, that helps expedite the learning process and ultimately solve a given task. Although this may seem like an appealing solution, it is often not obvious how to formulate an effective intrinsic reward. Recent works have introduced goal-oriented RL (Schaul et al., 2015; Chiang et al., 2019) as a way of constructing intrinsic rewards. However, most goal-oriented RL algorithms require large amounts of reward shaping (Chiang et al., 2019) or human demonstrations (Nair et al., 2018). **(C2)** The second issue is how to retrieve relevant past experiences that help to learn faster and improve sample efficiency. Recent attention has focused much on episodic RL (Blundell et al., 2016), which builds a non-parametric episodic memory to store past experiences, and thus can rapidly latch onto related ones through search with similarity. However, most episodic RL algorithms require additional memory space (Pritzel et al., 2017).

In this work, we propose Graph Structured Reinforcement Learning (GSRL), which constructs a state-transition graph, leverages structural graph information, and presents solutions for the problems raised above. As shown in Figure 1, when we encounter a complex task, we draw a map for planning with a long horizon (Eysenbach et al., 2019) and scan through the graph to seek related experiences within a short horizon (Lee et al., 2019).

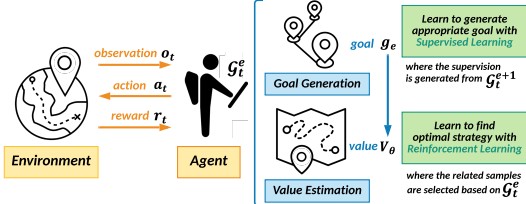

Figure 1: An illustration for motivation of GSRL.

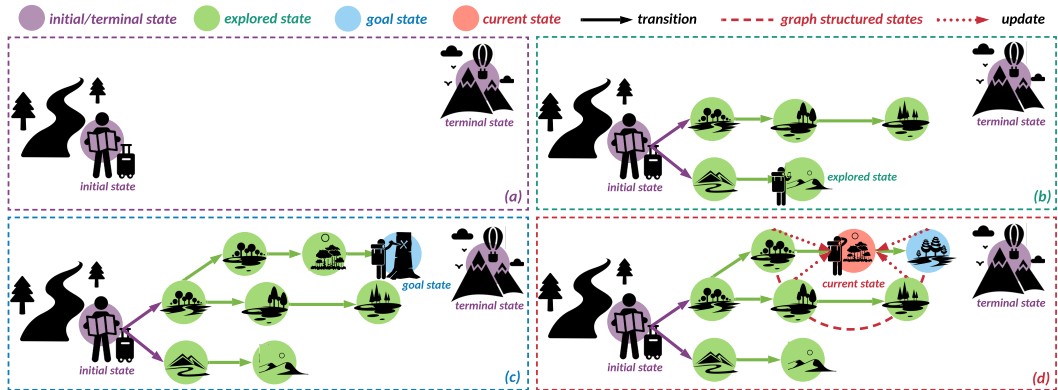

Figure 2: GSRL allows joint optimization of map construction and agent exploration from scratch. (a) An agent is located at the initial state and aims at the terminal state with an empty map at the beginning. (b) The agent explores the environment and records the explored states on the map. (c) The agent generates a goal from previous states to decompose the whole task into a sequence of easier tasks. (d) The agent updates current policy with related trajectories selected by structured map information.

Inspired by this, we propose a method to address the first problem **(C1)** by learning to generate an appropriate goal $g_e$ based on map $\mathcal{G}_0^e$ at episode $e$ and form the intrinsic rewards during goal generation. This goal generation can be optimized in hindsight. That is, we can update the goal generation model under the supervision generated by planning algorithms based on map $\mathcal{G}_0^{e+1}$ at episode $e + 1$. We can solve the second problem **(C2)** by updating the value estimation of the current state with the trajectories that include neighborhood states. Note that GSRL draws a map to help agents plan an appropriate direction in the long horizon and learn an optimal strategy in the short horizon, which can be widely incorporated with various learning strategies such as DQN (Mnih et al., 2013), DDPG (Lillicrap et al., 2015), etc.

However, arriving at a well-explored map in terms of state-transitions in the first place from scratch remains a challenging problem. In this regard, GSRL should be able to jointly optimize map construction, goal selection, and value estimation. As illustrated in Figure 2, we design GSRL mainly following four steps: (a) At the beginning, an agent is located at the initial state and aims at the terminal state with an empty map; (b) The agent then explores the environment with the current policy and records those explored states on the map; (c) Considering the far distance to the terminal state and sparse reward during the exploration, we decompose the whole task into a sequence of easier tasks guided by goals; (d) We leverage structured map information to select related trajectories to update the current policy; The agent can repeat procedures (b)-(d) and finally reach the terminal state with a well-explored map.

Our primary contribution is a novel algorithm that constructs a dynamic state-transition graph from scratch and leverages structured information to find an appropriate goal in goal selection with a long horizon and sample related trajectories in value estimation with a short horizon. This framework provides several benefits: (1) How to balance exploration and exploitation is a fundamental problem in any reinforcement learning (Andrychowicz et al., 2017; Nachum et al., 2018). GSRL learns an attention mechanism on the graph to decide whether to explore the states with high uncertainty or exploit the states with well estimated value. (2) Almost all the existing reinforcement learning approaches (Paul et al., 2019; Eysenbach et al., 2019) utilize randomly sampled trajectories to update the current policy leading to inefficiency. GSRL selects highly related trajectories for updates through the local structure in the graph. (3) Unlike previous approaches (Kipf et al., 2020; Shang et al., 2019), GSRL constructs a dynamic graph through agent exploration without any additional object representation techniques and thus is not limited to several specific environments. Empirically, we find that our method constructs a useful map and learns an efficient policy for highly structured tasks. Comparisons with state-of-the-art RL methods show that GSRL is substantially more successful for boosting the performance of a graph-structured RL algorithm.

## 2 PRELIMINARIES

Let $M = (S, A, T, P, R)$ be an Markov Decision Process (MDP) where $S$ is the state space, $A$ is the action space whose size is bounded by a constant, $T \in \mathbb{Z}_+$ is the episode length, $P : S \times A \rightarrow \Delta(S)$ is the transition function which takes a state-action pair and returns a distribution over states, and $R : S \times A \rightarrow \Delta(\mathbb{R})$ is the reward distribution. The rewards from the environment, called extrinsic,

are usually sparse and, therefore, difficult to learn. To address this, goal-oriented RL introduces goal $g \in G$ to build dense intrinsic rewards $R_g : S \times A \times G \to \Delta(\mathbb{R})$. In order to make the learning procedure stable, the goal is limited to be updated episodically. A policy $\pi : S \times G \to \Delta(A)$ prescribes a distribution over actions for each state and goal. At each timestep, the agent samples an action $a \sim \pi(s, g)$ and receives a corresponding reward $r_g(s, a)$ that indicates whether or not the agent has reached the goal. The episode terminates only after $T$ timesteps even when the agent reaches the goal. The agent's task is to maximize its cumulative discounted future reward. We use an off-policy algorithm to learn such a policy, as well as its associated goal-oriented $Q$-function and value function:

$$Q^{\pi}(s, a, g) = \mathbb{E}\left[\sum_{t=0}^{T-1} \gamma^t \cdot r_g(a_t, s_t) \mid s_t = s, a_t = a, \pi\right], \; V^*(s, g) = \max_a Q^*(s, a, g). \quad (1)$$

We use off-policy algorithms named DQN (Mnih et al., 2013) for discrete action space and DDPG (Lillicrap et al., 2015) for continuous action space to learn $Q$-functions and policies by utilizing off-policy data (i.e., data stored in the replay buffer).

## 3 GRAPH STRUCTURED REINFORCEMENT LEARNING

As illustrated in Figure 1, GSRL constructs a dynamic graph on top of state transitions. However, learning and planning algorithms on the state transition graph are non-trivial. In this section, we provide a theoretical analysis to show that exploration without any constraint would lead to the explosion of the graph. In order to guide the exploration with directions, we develop a novel goal-oriented RL framework which incorporates the structure information of the state-transition graph. Specifically, we first divide states into several groups according to their uncertainty and locations in the graph. We then adopt an attention mechanism to select an appropriate group and assign the state with the highest value in the graph as a goal to encourage further exploration. We also propose to update goal generation hindsightly and value estimation with related trajectories, to help RL agents learn efficiently.

### 3.1 EXPLORE WITH DYNAMIC GRAPH

We directly build a dynamic and directed graph using sample trajectories from the replay buffer. We denote the state-transition graph at timestep $t$ in episode $e$ by $\mathcal{G}_t^e = \langle \mathcal{V}_t^e, \mathcal{E}_t^e \rangle$ with relations $\mathcal{R}_t^e$, the node space $\mathcal{V}_t^e$ represents state space $S_{\mathcal{G}_t^e}$, edge space $\mathcal{E}_t^e$ represents action space $A_{\mathcal{G}_t^e}$, and relations $\mathcal{R}_t^e$ represents transition $P_{\mathcal{G}_t^e}$. We denote the graph's boundary by $\partial \mathcal{G}_t^e$, which consists of observed but not fully explored states. An illustrated example of these notations above is available at Figure 7 in Appendix A.

Let an agent start from an initial state $s_0$. Then $\mathcal{G}_t^e$ grows from a single node (i.e., $\mathcal{G}_0^0 = \{s_0\}$) and expands itself at each timestep, leading to the sequence $\{\{\mathcal{G}_t^1\}_{t=0}^{T-1}, \{\mathcal{G}_t^2\}_{t=0}^{T-1}, \ldots, \{\mathcal{G}_t^E\}_{t=0}^{T-1}\}$ where $T$ denotes episode length and $E$ is the number of episodes. Given that state-transition graph $\mathcal{G}_t^e$ is always a connected directed graph, we describe the expansion behavior as consecutive expansion, which means no jumping across neighborhood allowed. We denote the graph increment at timestep $t$ in episode $e$ as $\Delta \mathcal{G}_t^e$, and then we can ensure that $\mathcal{G}_t^e \cup \Delta \mathcal{G}_t^e = \mathcal{G}_{t+1}^e \subseteq \mathcal{G}_t^e \cup \partial \mathcal{G}_t^e$.

**Proposition 1.** *(adapted from (Xu et al., 2020)) We assume that the probability of degree of an arbitrary state in the whole state-transition graph $\mathcal{G}_{whole}$ being less than or equal to $d$ is larger than $p$ (i.e., $P(deg(s) \leq d) > p, \forall s \in S_{\mathcal{G}})$. We then consider a sequence of consecutively expanding sub-graphs $\{\{\mathcal{G}_t^1\}_{t=0}^{T-1}, \{\mathcal{G}_t^2\}_{t=0}^{T-1}, \ldots, \{\mathcal{G}_t^E\}_{t=0}^{T-1}\}$, starting with $\mathcal{G}_1^0 = \{s_0\}$, for all $t \geq 0, e \geq 1$. We can ensure that $P(|S_{\mathcal{G}_t^e}| \leq \epsilon) > p^{\epsilon}$, where $\epsilon = \frac{d \cdot (d-1)^{T \cdot e + t} - 2}{d - 2}$ when $d > 2$ and $\epsilon = 1, 3$ when $d = 1, 2$ respectively.*

*Proof.* The basic proof structure follows from (Xu et al., 2020); however, because our task and detailed setting are considerably different, several modifications are required as described in in Appendix D.1. □

The proposition implies that even if the given assumption of the whole state-transition graph $\mathcal{G}_{whole}$ has a small $d$ and a large $p$ (i.e., sparse graph), the guarantee of upper-bounding $|S_{\mathcal{G}_t^e}|$ becomes looser and weaker as $t, e$ get larger. In other words, randomly appending $\mathcal{G}_t^e$ through exploration would enhance computation complexity for learning and plan on the dynamic graph. In order to

prevent $\mathcal{G}_t^e$ from an explosion, we need to constrain $\Delta \mathcal{G}_t^e$ by guiding exploration with goals. In this paper, we propose to leverage the graph structure to assist agent exploration. Before the detailed introduction of exploration strategy, we firstly introduce the definition of certainty as

**Definition 1.** *Given a state $s$, we define its certainty as to the number of candidate actions that have been already taken. In this paper, we build the state-transition graph $\mathcal{G}$ on the replay buffer and thus can adopt the out degree* * of each state to approximately measure the certainty. In other words, we can approximate the certainty of state $cert(s) \approx deg(s)$, where $deg(s)$ denotes the degree of node $s$.*

Note that certainty of state in Definition 1 can be served as a local measurement to show the extent of exploration on some states, which is actually proportioned to the global measurement (i.e., the number of visited states) in a deterministic environment (See Appendix E.1 for details). In addition, we also can define uncertainty of state as the number of untaken candidate actions.

**Exploration Strategy.** A simple but effective way is to do exploration based on goal-oriented RL, which decomposes the whole task into a sequential of goal-oriented tasks. In this case, newly explored states in each episode are guided by a specific goal. Therefore, we constrain the state-transition graph from the explosion and balance the exploration and exploitation through assigning an appropriate goal to the agent. In order to further discuss the definition of an appropriate goal, we here provide the definition of the optimal goal. As shown in Figure 4, once we are accessible to the whole state-transition graph $\mathcal{G}_{\text{whole}}$, we can obtain the optimal solution by short path planning algorithm such as Dijkstra's Algorithm. We define the optimal goal as the terminal state on the path.

**Definition 2.** *For fully explored state-transition graph $\mathcal{G}_{full}$[†], we define the optimal goal $g_{full}^*$ as the terminal state (e.g., $s_{10}$ in Figure 4(a)). For any not fully explored state-transition graph $\mathcal{G}_0^e$, we define the optimal goal $g_e^*$ hindsightly, where we generate the optimal solution path $\mathcal{P}_{e+1}$ based on the shortest path planning on the next episode graph $\mathcal{G}_0^{e+1}$ and regard the reachable terminal state included both in $\mathcal{G}_0^e$ and $\mathcal{P}_{e+1}$ as $g_e^*$ (e.g., $s_6$ in Figure 4(c)).*

One should be noted that the optimal goal is hindsight generated and keep approaching the terminal state during exploration. We further analyze and discuss the relations between goals in this paper and previous goal-oriented RL literature in Appendix E.2.

**Proposition 2.** *Assume that $Q$-value of each state is well estimated (i.e., $Q = Q^*$), then optimal goal $g_e^*$ at the beginning timestep of any episode $e$ is always included in the boundary of the state-transition graph $\mathcal{G}_0^e$ (i.e., $\partial \mathcal{G}_0^e$).*

*Proof.* The proof of Proposition 2 can be found in Appendix D.2. □

The proposition implies that the candidate set for each goal generation can be limited to the boundary. However, learning to generate an appropriate goal is non-trivial with respect to a dynamic and large-scale state-transition graph $\mathcal{G}_t^e$. One intuitive solution is to divide the whole candidate goal space (i.e., $\partial \mathcal{G}_t^e$) into several candidate groups $\mathcal{C}_1, \ldots, \mathcal{C}_N$, where $N$ is the number of groups. In order to both cover all the

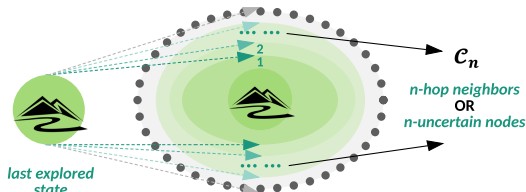

Figure 3: An illustration of exploration strategy.

states and control exploration size in these groups, our group segment should follow the principle that $\cup_{n=1}^N \mathcal{C}_n = \partial \mathcal{G}_t^e$ and $\mathcal{C}_m \cap \mathcal{C}_n = \emptyset$ for any $m \neq n; m, n = 1, 2, \ldots, N$.

Here, we set the last visited state $s_{\text{last}}$ in the previous episodes as $\mathcal{C}_0$, since $s_{\text{last}}$ often is both close to the terminal state and has high uncertainty. As shown in Figure 3, based on the last explored state, we build group $\mathcal{C}_n, n = 1, \ldots, N$ by extending scales in two different perspectives. We list two perspectives, as follows:

- *Extending from Neighbor Nodes.* One perspective is to leverage local structure on graph, where $\mathcal{C}_n := \{s\}, s \in \mathcal{N}^n(s_{\text{last}}) \cap \partial \mathcal{G}_t^e$ for $n = 1, 2, \ldots, N$, and $\mathcal{C}_N := \partial \mathcal{G}_t^e - \cup_{n=1}^{N-1} \mathcal{C}_n$. $\mathcal{N}^n(s_{\text{last}})$ means $n$-hop neighbors of $s_{\text{last}}$. The intuitive motivation behind this is to keep the learning procedure stable by slowly adjusting the goal to explore the surrounding environment.

---

*In this paper, we use 'degree' to represent 'out-degree', unless otherwise stated.

[†]We define the full explored graph $\mathcal{G}_{\text{full}}$ as the graph covering the optimal solution, and the whole graph $\mathcal{G}_{\text{whole}}$ as the graph covering all the state-transitions. Hence, we have that $\mathcal{G}_{\text{full}}$ is a sub-graph of $\mathcal{G}_{\text{whole}}$ (i.e., $\mathcal{G}_{\text{full}} \subseteq \mathcal{G}_{\text{whole}}$).

- *Extending from Uncertain Nodes.* The other perspective is to utilize certainty information to guide goal generation, where $\mathcal{C}_n := \{s\}, s \in S_{d=|A|-n} \cap \partial \mathcal{G}_t^e$ for $n = 1, 2, \ldots, N$, and $\mathcal{C}_N := \partial \mathcal{G}_t^e - \cup_{n=1}^N \mathcal{C}_n$. $S_{d=|A|-n}$ denotes set of states whose degree equals $|A| - n$, and $|A|$ is the size of action space. The intuitive motivation behind this is to eliminate uncertainty in the graph through exploration.

We provide the complexity analysis for two perspectives as follows. Let $d_{\partial \mathcal{G}_t^e}$ denote the maximum degree of states in $\partial \mathcal{G}_t^e$, and $|S_{\partial \mathcal{G}_t^e}|$ denote the number of states in $\partial \mathcal{G}_t^e$. The complexity to construct $\mathcal{C}_1, \ldots, \mathcal{C}_N$ by extending from neighbor nodes is $\mathcal{O}(d_{\partial \mathcal{G}_t^e}^{N-1})$ and by extending from uncertain nodes is $\mathcal{O}(|S_{\partial \mathcal{G}_t^e}|)$. The detailed analysis and method are available in Appendix E.3.

**Attention Strategy.** One should be noted that extending from either neighbor or uncertain nodes can guarantee the number of groups $\mathcal{C}_1, \ldots, \mathcal{C}_N$ on a dynamic graph $\mathcal{G}_t^e$ is fixed. In order to keep all the groups not empty, we append $s_{\text{last}}$ to them at the beginning timestep of each episode. However, RL algorithms face an exploration-exploitation dilemma, which requires the agent to makes appropriate selection rather than random exploration over these groups. In order to the trade-off between exploration and exploitation, we apply an attention mechanism to select an appropriate group on the current situation. For each group, we learn an embedding vector to represent its feature (i.e., certainty for *extending from uncertain nodes*). In this paper, the attention module runs over $N$ groups at each episode and select the appropriate one, which needs to take the features of other groups into consideration. Therefore, it's natural to adopt self-attention mechanism (Vaswani et al., 2017) here. For simplicity, we structure an embedding vector for each group and denote all the features of groups as $[f_1, \ldots, f_N]$, and then define $Q = K = V := (f_1, \ldots, f_N)^T \in \mathbb{R}^{N \times d}$. The self-attention can be defined as

$$ATT_\phi(\mathcal{C}_1, \ldots, \mathcal{C}_N) = \text{softmax}(\frac{QK^T}{\sqrt{N}})V, \tag{2}$$

where $ATT_\phi$ denotes self-attention function parameterized by $\phi$. The output of the self-attention is then fed to a multi-layer perception (MLP) with ReLU activation function. The output of MLP is in the dimension $\mathbb{R}^{N \times 1}$. And the selected group can be obtained according to

$$\mathcal{C}_{ATT} = \arg\max_{\mathcal{C}_n} \sigma(\text{MLP}(ATT_\phi(\mathcal{C}_1, \ldots, \mathcal{C}_N))), \tag{3}$$

where $\sigma(\cdot)$ denotes a sigmoid function, and $\mathcal{C}_{ATT}$ means the group selected according to the attention score. At the beginning timestep of any episode $e$, we have access to state-transition graph $\mathcal{G}_0^e$ and its boundary $\partial \mathcal{G}_0^e$. We then divide $\partial \mathcal{G}_0^e$ into groups $\mathcal{C}_1, \ldots, \mathcal{C}_N$, and obtain $\mathcal{C}_{ATT}$ through the aforementioned attention mechanism. We select one state with the highest value in $\mathcal{C}_{ATT}$ as the goal, which can be formulated as

$$g_e = \arg\max_s V(s, g_{e-1})^\ddagger, \ \forall s \in \mathcal{C}_{ATT}. \tag{4}$$

In this way, we can generate an appropriate goal $g_e$ to guide the agent exploration in the episode $e$.

## 3.2 LEARN WITH GRAPH STRUCTURED REINFORCEMENT LEARNING

As illustrated in Figure 1, we update goal generation under supervised learning and value estimation under reinforcement learning. Specifically, we introduce learning strategies as follows:

**Goal Learning Strategy.** As mentioned above, we learn the goal generation hindsight. As illustrated in Figure 4, at the beginning timestep of each episode $e + 1$, we first compute the shortest path distance between initial state ($s_0$ in (c)) and highest value state ($s_9$ in (c)) to obtain the solution path ($\mathcal{P}_{e+1} = \langle s_0, s_1, s_3, s_6, s_9 \rangle$ in (d)). We then find the optimal goal $g_e^{*\S}$ at episode $e$ based on the solution path generated at episode $e + 1$. Specifically, we search for the intersection state between $\mathcal{P}_{e+1}$ and $\partial \mathcal{G}_t^e$ in the inverse order of the solution path. Given that $s_9$ is unexplored and unreachable in the episode $e$, we select $s_6$ as the optimal goal $g_e^*$. We then find the optimal group $\mathcal{C}^*$ that contains goal $g_e^*$. With this hindsight supervision on the group selection, we are able to update the attention

---

‡Similar as (Andrychowicz et al., 2017), we relabel all the rewards when the goal changes. Therefore, $V(s, g_{e-1})$ is used here.

§Note that we define the target state and the optimal solution path in episode $e + 1$, and the optimal goal in episode $e$.

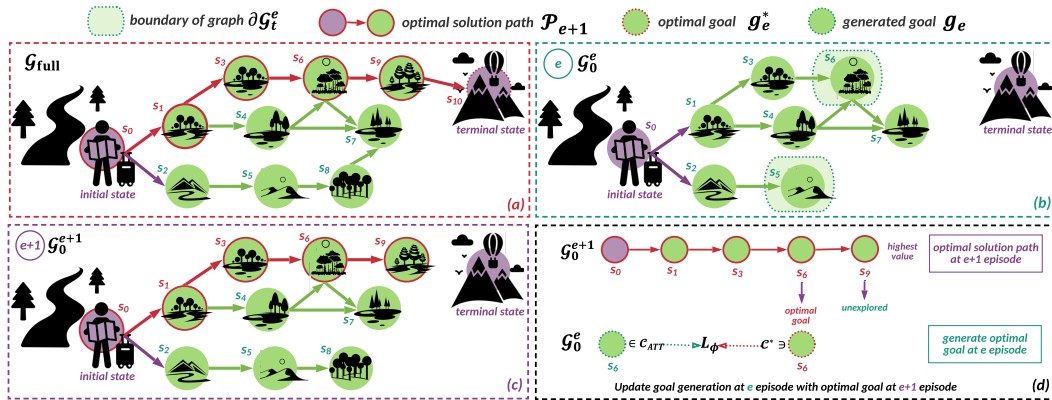

Figure 4: Illustrations of optimal goals. States surrounded by red circles and connected by red arrows are optimal solution path, which can be obtained by the shortest path planning algorithm on the fully explored graph $\mathcal{G}_{\text{full}}$ in (a). When the graph is not fully explored ($\mathcal{G}_0^e$ in (b)), we generate the optimal solution path hindsight, where we regard the state with the highest value in the next episode ($\mathcal{G}_0^{e+1}$ in (c)) as the target state, and shortest path to it as the optimal solution path ($\mathcal{P}_{e+1} = \langle s_0, s_1, s_3, s_6, s_9 \rangle$ in (d)). We define the optimal goal $g_e^*$ as the reachable target state in the optimal solution path in episode $e$ (e.g., $s_{10}$ in $\mathcal{G}_{\text{full}}$ and $s_6$ in $\mathcal{G}_0^e$).

mechanism in Eq. (3) via a standard supervised learning algorithm, where the objective function can be formulated as

$$L_\phi = \mathbb{E}_{(s,a,g,s',r) \sim \mathcal{D}} \left[ (\mathcal{C}^* - \mathcal{C}_{ATT})^2 + \alpha \cdot \|\phi\|_2 \right], \tag{5}$$

where $\|\phi\|_2$ is regularizer and $\alpha$ is the corresponding hyper-parameter. This method updating goal generation under the supervision of the group instead of goal can eliminate instability brought from potentially inaccurate value estimation because our group division does not involve the result from value estimation.

**Value Learning Strategy.** With a generated goal at the beginning of each episode, we can build the critical block of our method, i.e., a goal-conditioned policy and its associated value function. We consider a goal-reaching agent interacting with an environment. The agent observes its current state $s \in S$ and a goal state $g \in G$. The dynamics are governed by the distribution $P(s_{t+1}|s_t, a_t)$. At every timestep, the agent samples an action $a \sim \pi(a|s, g)$ and receives a corresponding reward $r_g(s, a)$ that indicates whether the agent has reached the goal. The episode terminates after $T$ timesteps, and even the agent reaches the goal. The agent's task is to maximize its cumulative and undiscounted reward. We use an off-policy algorithm to learn such a policy, as well as its associated goal-conditioned $Q$-function. For example, we obtain a policy by acting greedily, *w.r.t.*, the $Q$-function as

$$Q(s, a, g) \leftarrow r_g(s, a) + \gamma \cdot \max_{a'} Q(s, a', g). \tag{6}$$

In order to improve data efficiency and obtain good value estimation, we consider choosing an off-policy RL algorithm with goal relabelling and update parameters with related trajectories. Specifically, we choose an off-policy RL algorithm with goal relabelling, such as Andrychowicz et al. (2017). We define related data as those trajectories that contain neighborhood nodes of the current state. Formally, when we update $Q$-function of state $s$, we sample $\mathcal{D}_{\text{related}} := \{\tau\}$ from replay buffer $\mathcal{D}$, where $\tau$ denotes the trajectory that contains at least one state in $s$'s neighborhood $\mathcal{N}^1(s)$. The $Q$-network is learned by minimizing the following objective function:

$$L_\theta = \mathbb{E}_{(s,a,g,s',r) \sim \mathcal{D}_{\text{related}}} \left[ (r_g + \gamma \cdot \max_a Q_\theta(s', a, g) - Q_\theta(s, a, g))^2 + \beta \cdot \|\theta\|_2 \right], \tag{7}$$

where $\beta$ is the weight of the regularization term. Besides, as we show in Proposition 3, our GSRL algorithm can converge to a unique optimal point if the $Q$-learning strategy is adopted.

**Proposition 3.** *Denote the Bellman backup operator in Eq. (6) as $\mathcal{B} : \mathbb{R}^{|S| \times |A| \times |G|} \to \mathbb{R}^{|S| \times |A| \times |G|}$ and a mapping $Q : S \times A \times G \to \mathbb{R}^{|S| \times |A| \times |G|}$ with $|S| < \infty$ and $|A| < \infty$. Repeated applications of the operator $\mathcal{B}$ for our graph-based state-action value estimate $\hat{Q}_\mathcal{G}$ converges to a unique optimal value $\hat{Q}_{\mathcal{G}^*}^*$ with well-explored graph $\mathcal{G}^*$ (i.e., fully explored graph $\mathcal{G}_{\text{full}}$).*

*Proof.* The proof is shown in Appendix D.3. ☐

**Overall Algorithm.** We provide the overall algorithm in Algorithm 1 in Appendix B, and an illustrated example in Figure 8 in Appendix C. One can see that our state-transition graph can be

applied to both the learning and planning phase. Previous graph-based RL algorithms either focus on learning (Zhu et al., 2019) or planning (Eysenbach et al., 2019) based on structured information of the state-transition graph.

# 4 EXPERIMENTS

In this section, we perform an experimental evaluation of our algorithm and compare it with other state-of-the-art methods. Our experiment environments are based on the standard robotic manipulation environments in the OpenAI Gym (Brockman et al., 2016). We provide the experimental results to answer the following questions:

1. Can GSRL obtain better convergence in various environments?
2. Can GSRL tackle a high-dimensional and continuous environment with obstacles?
3. Can GSRL perform higher sample efficiency?
4. What is the influence of group selection for goal generation and related experience for value estimation in GSRL?

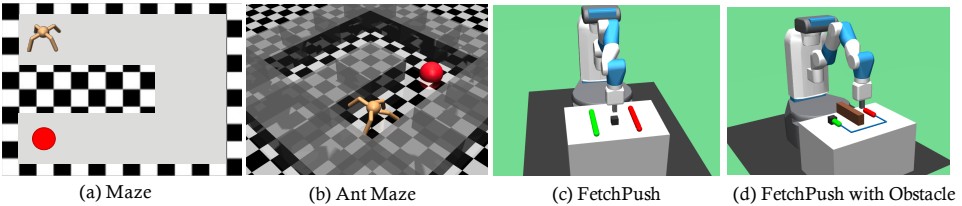

|  (a) Maze | (b) Ant Maze | (c) FetchPush | (d) FetchPush with Obstacle |

Figure 5: Visualization of robotic manipulation environments.

In order to answer the first question, we demonstrate our method in various robotic manipulation tasks including reach task (see Figure 5(a)) and fetch task (see Figure 5(c)). To answer the second one, we investigate how our method performs on environment with high-dimensional state and action space (see Figure 5(b)) and obstacle (see Figure 5(d)). We answer the third one by comparison on the convergence of learning curves in Figure 6. By sample efficiency, we further analysis exploration performance in Appendix F.5. Specifically, we conduct experiments with previous approaches:

- **HER**: Andrychowicz et al. (2017) generated imaginary goals in a simple heuristic way to tackle the sparse reward issue.
- **MAP**: Huang et al. (2019) explicitly modeled the environment in a hierarchical manner, with a high-level map abstracting the state space and a low-level value network to derive local decisions.
- **GoalGAN**: Florensa et al. (2018) leveraged Least-Squares GAN (Mao et al., 2018b) to mimic the set of Goals of Intermediate Difficulty as an automatic goal generator.
- **CHER**: Fang et al. (2019) proposed to enforce more curiosity in earlier stages and changes to larger goal-proximity later.

One should be noted that graph structure in GSRL is constructed on top of the replay buffer for goal generation and value estimation, which can be closely incorporated with policy networks such as DQN (Mnih et al., 2013), DDPG (Lillicrap et al., 2015), etc. To demonstrate the real performance gain from our GSRL, we set the policy network with DDPG for GSRL and all the baselines. The detailed description of environments, experiment settings, and implementation details can be found in Appendix F.1.

**Maze**. We first test our method and other strong baselines in the Maze environment, where the ant agent learns to reach the specified target position ($\epsilon$-ball depicted in red) located at the other end of the U-turn as shown in Figure 5(a). This environment is quite simple, where most RL approaches can converge with a success rate 1. As Figure 6(a) illustrates, GSRL performs with high sample efficiency.

**AntMaze**. We show that our GSRL is efficient in a complex environment of robotic agent navigation tasks, as illustrated in Figure 5(b), where the state space is of 8 dimensions and the action space is of 30 dimensions. Duan et al. (2016) showed that standard RL methods are unable to solve it. As shown in Figure 6(b), GSRL achieves the best asymptotic performance and sample efficiency.

**FetchPush**. As shown in Figure 5(c), in the fetch environment, the agent is trained to fetch an object from the initial position (rectangle in green) to a distant position (rectangle in red). Although

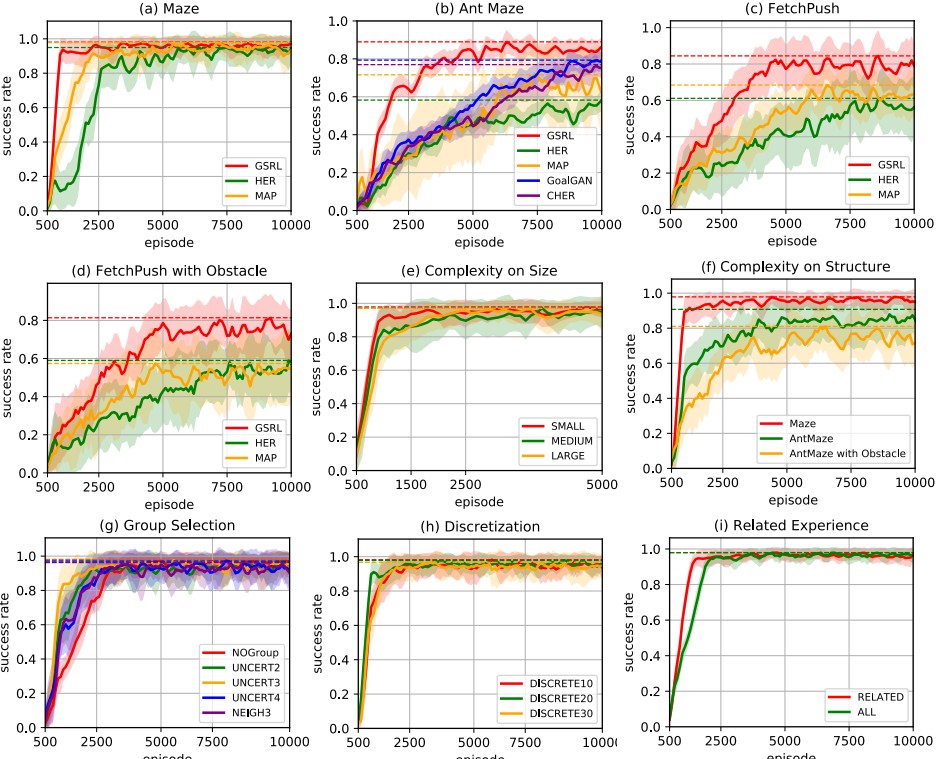

Figure 6: Learning curves of GSRL, HER, MAP, GoalGAN, and CHER on various environments with 10 random seeds, where the solid curves depict the mean, the shaded areas indicate the standard deviation, and dashed horizontal lines show the asymptotic performance.

the fetch tasks are more complicated than they reach ones in the maze, GSRL also yields large performance gain, as shown in Figure 6(c).

**FetchPush with Obstacle**. As Figure 5(d) illustrates, we create an environment based on FetchPush with a rigid obstacle, where the brown block is a static wall that cannot be moved. Experimental results in Figure 6(d) suggests that the graph structure of state-transitions can provide additional useful information, which leads to better guidance for goal generation and value estimation.

**Impact of Environment Size**. In order to investigate whether GSRL can be well adapted in the environments with different complexity sizes, we extend the Maze environment, as shown in Figure 5(a). Maze environment with larger size usually means more sparse rewards. We report the performance comparisons on the three environments with **SMALL**, **MEDIUM**, and **LARGE** sizes in Figure 6(e). More details of these environments are available in Appendix F.1. One can see that GSRL can be well address sparse reward issue in various environments.

**Impact of Environment Complexity**. In order to investigate whether GSRL can be well adapted in the environments with different complexity levels, we extend the Maze environment with a more complex or high-dimensional structure. We create **AntMaze with Obstacle** environment, where we extend the AntMaze environment with an obstacle. The performance of GSRL in this environment is illustrated in Figure 6(f). More details of these environments are available in Appendix F.1. One can see that GSRL can be well adapted in various environments with different complexity levels.

**Impact of Group Selection**. We provide two strategies to divide the boundary of the graph into several groups, namely *extending from neighbor nodes* and *extending from uncertain nodes* in Section 3. We demonstrate the performance of these two strategies in Figure 6(g). We set GSRL without using the attention strategy on the groups as **NOGroup**. We adopt the first strategy and set the number of groups as 3, which corresponds to **NEIGH3**. We then show the performance of the second strategy with the number of groups equal to 2, 3 and 4, which correspond to **UNCERT2**, **UNCERT3** and **UNCERT4**. Results show that GSRL without any group selection would lead to inefficiency since the goal generation strategy at the beginning is almost random. Both of these two strategies perform well, as illustrated. We also provide complexity analysis in Appendix E.3. In the main experiment, we adopt *extending from uncertain nodes* with 3 groups (i.e., UNCERT3).

**Impact of Discretization**. We apply $K$-bins discretization technique (Kotsiantis & Kanellopoulos, 2006) to discretize the continuous state and action spaces, where there is a wrapper that converts a

Box observation into a single integer. **DISCRETE10**, **DISCRETE20** and **DISCRETE30** in Figure 6(h) denote the performances of $K = 10$, 20 and 30 respectively. We find that for the simple tasks, the choice of $K$ is not critical. $K$ is set at 20 in the main experiment.

**Impact of Related Experience**. We further study the impact of using neighborhood structure of the state-transition graph to select related experience to efficiently update value estimation. We build a variant to update value estimation from transitions drawn uniformly from the replay buffer $\mathcal{D}$, denoted as **ALL**. **RELATED** is used to denote GSRL, where $\mathcal{D}_{\text{related}}$ is adopted instead of $\mathcal{D}$.

## 5 RELATED WORK

**Structured Models of Environments**. Recent work has investigated leveraging structured environments to make great strides in improving predictive accuracy (Kipf et al., 2018; Xu et al., 2019; Chang et al., 2017; Battaglia et al., 2016; Watters et al., 2017; Sanchez-Gonzalez et al., 2018) and accelerating reinforcement learning procedures (Shang et al., 2019; Kipf et al., 2020; Eysenbach et al., 2019; Wang et al., 2018). These methods mainly first construct some form of graph neural network where node update functions model the dynamics of individual objects, parts, or agents and edge update functions model their interactions and relations through unsupervised object discovery (Xu et al., 2019), contrastive learning (Kipf et al., 2020) or agent exploration (Eysenbach et al., 2019). Unlike these previous works based on well-organized graphs, our model GSRL constructs and makes use of the graph from scratch. Specially, GSRL constructs a dynamic map via exploration and builds efficient exploration with structured information from this map, which allows our approach to be widely deployed into more complex environments.

**Goal-oriented Reinforcement Learning.** Goal-oriented RL allows an agent to generate intrinsic rewards, which is defined with respect to target subsets of the state space called goals (Florensa et al., 2018; Paul et al., 2019). Recently, goal-oriented RL has been investigated widely in various deep RL scenarios such as imitation learning (Pathak et al., 2018; Srinivas et al., 2018), disentangling task knowledge from the environment (Mao et al., 2018a; Ghosh et al., 2019), constituting lower-level controller in hierarchical RL (Shang et al., 2019; Nachum et al., 2018) and organizing cooperation in multi-agent RL (Jin et al., 2019; Yang et al., 2020). Hence, how to generate appropriate goals is the essential technique in any goal-oriented RL (Andrychowicz et al., 2017; Ren et al., 2019). Eysenbach et al. (2019) proposed to utilize the shortest-path search on replay buffer to generate a sequence of goals. Instead of planning goals from a well-explored state-transition graph, in this paper, we investigate guiding exploration with structured information from a dynamic graph, which can largely improve the performance in terms of sample efficiency in tasks with sparse reward signals. (Zhao et al., 2019) proposed to maximize entropy of selected trajectories; however, our method utilizes the structure information in the state-transition graph to select related trajectories for learning.

**Hierarchical Reinforcement Learning**. Hierarchical RL learns a set of primitive tasks that together help an agent learn the complex task. There are mainly two lines of work. One class of algorithms (Shang et al., 2019; Nachum et al., 2018; Bacon et al., 2017; Vezhnevets et al., 2017) jointly learn a low-level policy together with a high-level policy, where the lower-level policy interacts directly with the environment to achieve each task, while the higher-level policy instructs the lower-level policy via high-level actions or goals to sequence these tasks into the complex task. The other class of methods (Bagaria & Konidaris, 2019; Fox et al., 2017; Hartikainen et al., 2019; Pitis et al., 2020; Pong et al., 2019) focus on discovering sub-goals that are easy to reach in a short time, or options which are lower-level control primitives, can be invoked by the meta-control policy. The common idea GSRL shares with Hierarchical RL is to decompose the complex task into several sub-tasks to achieve. In this paper, the key difference is that, we propose to build the state-transition graph and utilize the structure information for goal generation and value estimation.

## 6 CONCLUSION

In this paper, we propose a novel framework called GSRL, which leverages structure information of the state-transition graph for efficient goal generation and value estimation. We provide theoretical analysis to show the efficiency and converge property of our method. Results on various challenging robotic manipulation environments demonstrate that GSRL can outperform the state-of-the-art RL algorithms. In the future, this work would shed light on graph-structured RL for efficient learning and planning, where various graph-based algorithms can help RL agents learn and infer in highly structured environments.

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

## A    NOTATIONS

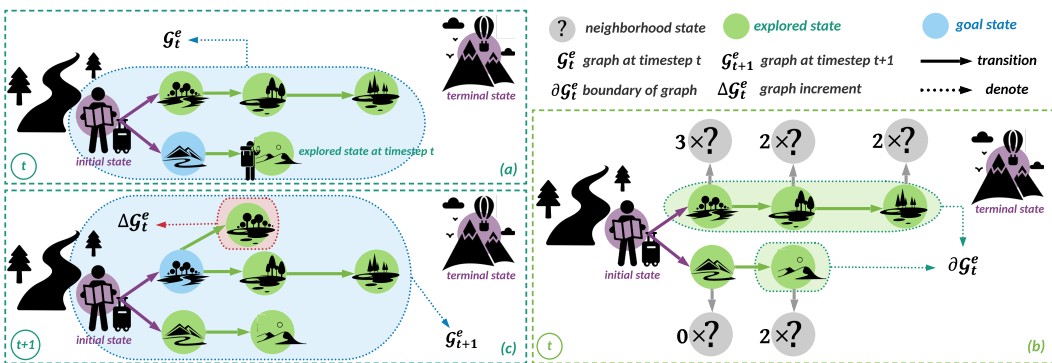

Figure 7: An illustrated example of notations in Graph Structured Reinforcement Learning (GSRL).

In this paper, we construct a state-transition graph on top of the replay buffer. As Figure 7(a) shows, we build the graph $\mathcal{G}_t^e$ based on historical explored trajectories at timestep $t$ in episode $e$. For any not fully explored state-transition graph, there exist many poorly-explored states. We measure the exploration (i.e., certainty in Defintion 1) of these states according to the number of their untaken candidate actions. As illustrated in Figure 7(b), we define the boundary of $\mathcal{G}_t^e$ as a set of states, at least one of whose candidate actions is not readily taken. Each untaken action may lead to unvisited states (denoted by ? icon). We denote the boundary as $\partial \mathcal{G}_t^e$. As illustrated in Figure 7(c), after each timestep $t + 1$, the agent explored a new state denoted as $\Delta \mathcal{G}_t^e$. Then, $\mathcal{G}_t^e$ and $\Delta \mathcal{G}_t^e$ together make up the dynamic graph at timestep $t + 1$ denoted as $\mathcal{G}_{t+1}^e$.

## B    ALGORITHM

---

**Algorithm 1** Graph Structured Reinforcement Learning (GSRL)

---

1: Initialize replay buffer $\mathcal{D} = \{s_0\}$ and state-trainsition graph $\mathcal{G} = \{s_0\}$
2: **for** epsiode number $e = 1, 2, \ldots, E$ **do**
3:     Select an appropriate group for exploration according to Eq. (3)
4:     Generate goal $g_e$ according to Eq. (4)
5:     **for** timestep $t = 0, 1, 2, \ldots, T - 1$ **do**
6:         Receive observation $s_t$ from environment
7:         $a_t \leftarrow \epsilon$-greedy policy based on $Q(s_t, a, g)$
8:         Take action $a_t$, receive reward $r_t$ and next state $s_{t+1}$
9:         Append $(s_t, a_t, r_t, s_{t+1}, g_e)$ to $\mathcal{D}$
10:        Relabel rewards $r_g$ with $g_e$
11:        Append $(s_t, a_t, s_{t+1})$ to $\mathcal{G}$ if $(s_t, a_t, s_{t+1}) \notin \mathcal{G}$
12:        **if** $t$ mod update_interval $== 0$ **then**
13:            Sample related experience $(s_t, a_t, s_{t+1}, r_t, g_e)$ to $\mathcal{D}_{\text{related}}$
14:            Update parameter $\theta$ using Eq. (7)
15:        **end if**
16:    **end for**
17:    Compute optimal goal $g_e^*$ according to Definition 2
18:    Update parameter $\phi$ using Eq. (5)
19: **end for**

---

We provide the overall algorithm in Algorithm 1. The key contribution of our paper is to leverage structured information in the state-transition graph for efficient goal generation and value estimation, which is represented in line 4 and 13, respectively. We then describe the overall procedure of GSRL according to Algorithm 1 as follows:

There is no graph structure for the agent to support when the task starts. Hence, the agent initializes the replay buffer $\mathcal{D}$ and the state-transition graph $\mathcal{G}$ in line 1. At the beginning timestep of each

episode $e$, we divide the boundary of the state-transition graph $\partial \mathcal{G}_0^e$ into $N$ groups and adopt attention mechanism to select an appropriate one for exploration in line 3. Within the selected group, we choose the state with the highest value as the generated goal $g_e$ in line 4. The agent tries to reach the goal state through current policy-based $Q$ value in line 7 and record interaction history in the replay buffer in line 9. As goal-oriented RL provides the agent intrinsic reward conditioned on the current goal, the agent is required to relabel reward with $r_g$ conditioned on $g_e$ in line 10. Then the agent updates the state-transition graph in line 11. In order to efficiently update policy, the agent sample related trajectories that contain at least one neighbor states of the current state in line 13. In line 14, the policy is updated with DDPG (Lillicrap et al., 2015). At the end of each episode $e$, the state-transition graph is actually built for episode $e + 1$ denoted as $\mathcal{G}_0^{e+1}$. The agent is able to find optimal goal $g_e^*$ through planning algorithm on $\mathcal{G}_0^{e+1}$ in line 17. The attention mechanism is updated by supervised learning in line 18.

## C  ILLUSTRATION

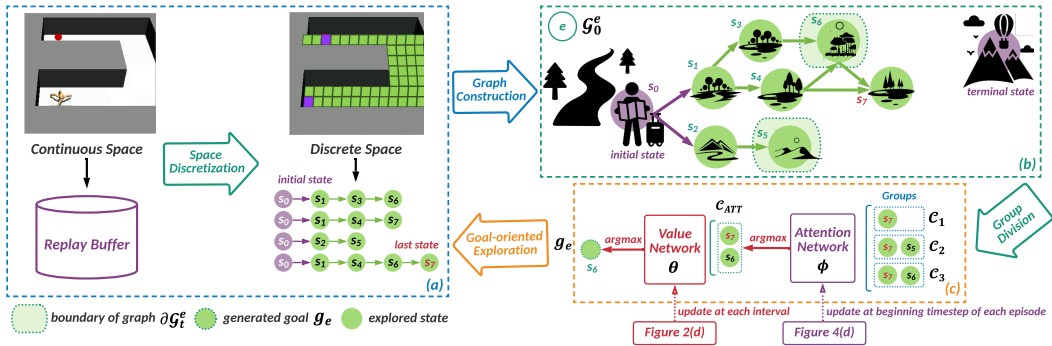

Figure 8: An illustrated example of Graph Structured Reinforcement Learning (GSRL).

We provide an illustrated example for GSRL in Figure 8. When an agent encounters a complex environment, GSRL first discretize the continuous space into a discrete one, as illustrated in (a). We color the last state (i.e., $s_{\text{last}} = s_7$) visited in the previous episodes with red. Meanwhile, the historical trajectories stored in the replay buffer can be represented with several paths (e.g., $\langle s_0, s_1, s_3, s_6 \rangle$). With these paths, we can easily construct a state-transition graph at the beginning timestep of episode (i.e., $\mathcal{G}_0^e$). As illustrated in (b), we can find the boundary of the graph (i.e., $\partial \mathcal{G}_t^e$) as shown in Appendix A. In this case, $s_5$, $s_6$ are in the boundary of the current graph (i.e., $\partial \mathcal{G}_0^e$). We then follow *extending from uncertain nodes* to divide the boundary into several groups. The number groups is the hyper-parameter, and we choose 3 here. For each group, we include $s_{\text{last}}$ since $s_{\text{last}}$ is often with high value and close to the terminal state. As shown in (c), we then adopt attention mechanism to select one group $\mathcal{C}_{ATT}$ and assign the state with the highest value in the select group as the goal $g_e$. With the generated goal, the agent can perform goal-oriented exploration to divide the complex task into several sub-tasks to deal with. There are two learned parts in GSRL, namely attention network for goal generation, and value network for value estimation. We design a hindsight learning approach to update the attention network at the beginning timestep of each episode (as shown in Figure 4), and use related experiences to update value network at each update interval (as shown in Figure 2).

## D  PROOFS

### D.1  PROOF OF PROPOSITION 1

**Proposition 1.** *(adopted from (Xu et al., 2020)) We assume that the probability of degree of an arbitrary state in the whole state-transition graph $\mathcal{G}_{whole}$ being less than or equal to $d$ is larger than $p$ (i.e., $P(deg(s) \leq d) > p, \forall s \in S_{\mathcal{G}}$). We then consider a sequence of consecutively expanding sub-graphs $\{\{\mathcal{G}_t^1\}_{t=0}^{T-1}, \{\mathcal{G}_t^2\}_{t=0}^{T-1}, \dots, \{\mathcal{G}_t^E\}_{t=0}^{T-1}\}$, starting with $\mathcal{G}_1^0 = \{s_0\}$, for all $t \geq 0, e \geq 1$. We can ensure that $P\left(|S_{\mathcal{G}_t^e}| \leq \epsilon\right) > p^\epsilon$, where $\epsilon = \frac{d \cdot (d-1)^{T \cdot e + t} - 2}{d - 2}$ when $d > 2$ and $\epsilon = 1, 3$ when $d = 1, 2$ respectively.*

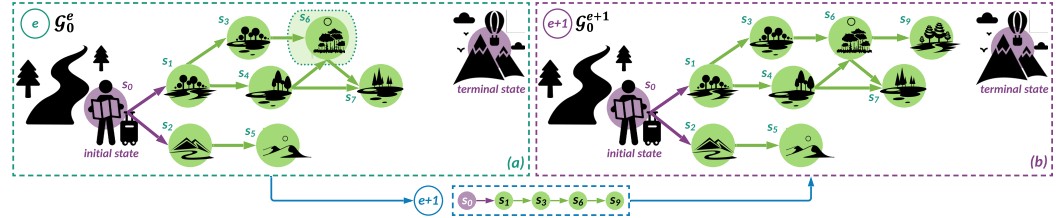

Figure 9: An illustrated example of the proof of Proposition 1 and the key difference between the proposition in this paper and the one in the previous work.

*Proof.* In (Xu et al., 2020), the new node of the current graph (i.e., $\mathcal{G}_0^e$) can be sampled directly from the neighborhood. As illustrated in Figure 9, considering the graph in this paper is the state-transition graph, we can not directly sample from the neighborhood. We only can obtain the new states by the explored trajectories (e.g., $\langle s_0, s_1, s_3, s_6, s_9 \rangle$ in Figure 9). Therefore, we propose to consecutively expand the sub-graphs at the timestep level (i.e., $\{\{\mathcal{G}_t^1\}_{t=0}^{T-1}, \{\mathcal{G}_t^2\}_{t=0}^{T-1}, \ldots, \{\mathcal{G}_t^E\}_{t=0}^{T-1}\}$, starting with $\mathcal{G}_1^0 = \{s_0\}$). Although we follow the framework in (Xu et al., 2020), several modifications are required. And, we provide the detailed proof as follows:

We consider the extreme case of greedy consecutive expansion at each timestep $t$ in any episode $e$, where $\mathcal{G}_{t+1}^e = \mathcal{G}_t^e \cup \Delta\mathcal{G}_t^e = \mathcal{G}_t^e \cup \partial\mathcal{G}_t^e$, since if this case satisfies the inequality, any case of consecutive expansion can also satisfy it. By definition, all the subgraphs $\mathcal{G}_t^e$ are a connected graph. Here, we use $\Delta S^t$ to denote $S_{\Delta\mathcal{G}^t}$ for short. In each episode, we can ensure that the newly added nodes $\Delta S_t^e$ at timestep $t$ only belong to the neighborhood of the last added nodes $\Delta S_{t-1}^e$.

Within each episode $e$, we study the sequence $\{\Delta\mathcal{G}_0^e, \Delta\mathcal{G}_1^e, \ldots, \Delta\mathcal{G}_{T-1}^e\}$, where $T$ is the episode length. In this case, each node in $\Delta S_t^e$ already has at least one edge within $\Delta\mathcal{G}_{t-1}^e$ due to the definition of connected graphs. We can have

$$P\big(|\Delta S_t^e| \leq |\Delta S_{t-1}^e| \cdot (d-1)\big) > p^{|\Delta S_{t-1}^e|}. \tag{8}$$

For $e = 1$ and $t = 0$, we have $P(|\Delta S_1^1| \leq d) > p$ and thus

$$P(|S_{\mathcal{G}_0^1}| \leq 1 + d) > p. \tag{9}$$

For $e \geq 1$ and $t \geq 1$, we analyze the consecutive expansion of the state-transition graph $\mathcal{G}$ as

$$\mathcal{G}^1 \rightarrow \mathcal{G}^2 \rightarrow \cdots \rightarrow \mathcal{G}^E$$
$$\Rightarrow \underbrace{\mathcal{G}_0^1 \rightarrow \mathcal{G}_1^1 \rightarrow \cdots \rightarrow \mathcal{G}_{T-1}^1}_{\mathcal{G}^1} \rightarrow \underbrace{\mathcal{G}_0^2 \rightarrow \mathcal{G}_1^2 \rightarrow \cdots \rightarrow \mathcal{G}_{T-1}^2}_{\mathcal{G}^2} \rightarrow \cdots \rightarrow \underbrace{\mathcal{G}_0^E \rightarrow \mathcal{G}_1^E \rightarrow \cdots \rightarrow \mathcal{G}_{T-1}^E}_{\mathcal{G}^E}. \tag{10}$$

Given that $|\Delta S_{\mathcal{G}_t^e}| \geq 1, \forall t \in [0, T-1]$, we consider the extreme case that $|\Delta S_{\mathcal{G}_t^e}| = 1, \forall t \in [0, T-1]$, which means that every exploration will result in a new explored state and should be respondes to the upper bound of the explosion. Based on $|\Delta S_{\mathcal{G}_t^e}| = 1 + |\Delta S_{\mathcal{G}_0^1}| + |\Delta S_{\mathcal{G}_1^1}| + \cdots + |\Delta S_{\mathcal{G}_{T-1}^1}| + |\Delta S_{\mathcal{G}_0^2}| + |\Delta S_{\mathcal{G}_1^2}| + \cdots + |\Delta S_{\mathcal{G}_{T-1}^2}| + \cdots + |\Delta S_{\mathcal{G}_0^e}| + \cdots + |\Delta S_{\mathcal{G}_t^e}|$, we have

$$P\big(|S_{\mathcal{G}^e}| \leq 1 + d + d \cdot (d-1) + \cdots + d \cdot (d-1)^{e \cdot T + t - 1}\big) > p^{1 + d + d \cdot (d-1) + \cdots + d \cdot (d-1)^{e \cdot T + t - 2}}. \tag{11}$$

When $d = 1$, there can be only one node, so in this case, $\epsilon = 1$. When $d = 2$, we follow Eq. (11) and derive that in this case, $\epsilon = 3$. When $d > 2$, it holds that

$$P\big(|S_{\mathcal{G}^t}| \leq \frac{d \cdot (d-1)^{e \cdot T + t} - 2}{d - 2}\big) > p^{\frac{d \cdot (d-1)^{e \cdot T + t - 1} - 2}{d - 2}}. \tag{12}$$

We can find that $t = 0$ also satisfies this inequality. □

## D.2   PROOF OF PROPOSITION 2

**Proposition 2.** *Assume that Q-value of each state is well estimated (i.e., $Q = Q^*$), then optimal goal $g_e^*$ at the beginning timestep of any episode $e$ is always included in the boundary of the state-transition graph $\mathcal{G}_0^e$ (i.e., $\partial\mathcal{G}_0^e$).*

*Proof.* According to Definition 2, as shown in Figure 10(a), in the fully explored graph $\mathcal{G}_{\text{full}}$, the optimal goal $g_{\text{full}}^*$ is the terminal state in the optimal solution $\mathcal{P}_{\text{full}}$, which is also the terminal state

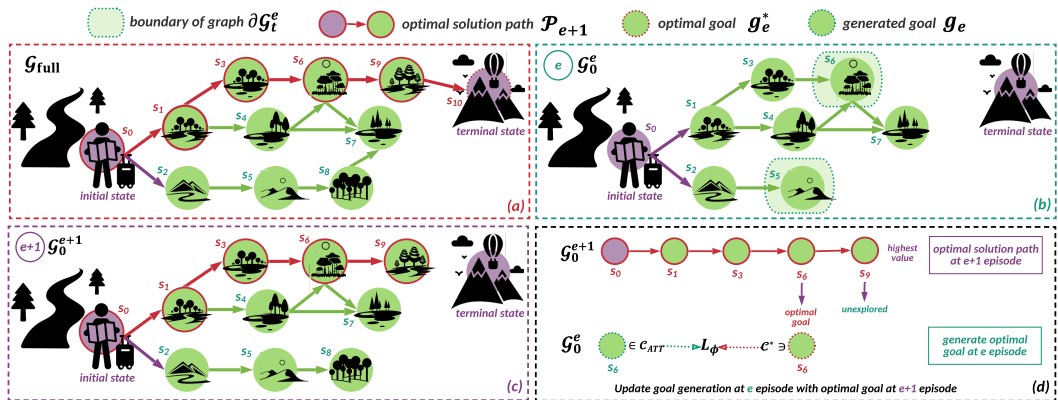

Figure 10: An illustrated example of the relationship between optimal goal and boundary. In the fully explored graph $\mathcal{G}_{\text{full}}$, the red circled states together show the optimal solution path ($\mathcal{P}_{\text{full}} = \langle s_0, s_1, s_3, s_6, s_9, s_{10} \rangle$) with terminal one ($s_{10}$) for the optimal goal in (a). In any other not fully explored state-transition graph $\mathcal{G}_0^e$ at the beginning timestep of any episode $e$ in (b), we regard the reachable state in the dashed line circle ($s_6$) through planning in the next episode $\mathcal{G}_0^{e+1}$ in (c) as the optimal goal in (d).

in the environment (i.e., $s_{10}$). The intuitive explanation behind this is very natural, where the environment in this case is fully explored, and thus the agent is ready to target at the terminal state. In the other cases, we generate the optimal goal $g_e^*$ of episode $e$ at the episode $e + 1$. Specially, we find the shortest path to the highest value state in $\mathcal{G}_0^{e+1}$ as the optimal solution path $\mathcal{P}_{e+1}$. As Figure 10 illustrates, in the episode $e + 1$, the highest value is $s_9$ and the optimal solution path in this case is $\mathcal{P}_{e+1} = \langle s_0, s_1, s_3, s_6, s_9 \rangle$. We then compare the explored states in $\mathcal{G}_0^e$ with the states in $\mathcal{P}_{e+1}^{\text{inverse}}$, where $\mathcal{P}_{e+1}^{\text{inverse}} = \langle s_9, s_6, s_3, s_1, s_0 \rangle$ is the inverse order of $\mathcal{P}_{e+1}$. As Figure 10(d) shows, finally we obtain $s_6$ as the optimal goal $g_e^*$. As stated above, it's easy to find that there are two cases in the optimal goal generation. One is the last node of solution path $\mathcal{P}_{e+1}$. The other is one of the rest nodes in $\mathcal{P}_{e+1}$ except the last one. We then prove that in both of these cases, optimal goal $g_e^*$ is always included in the boundary of the state-transition graph $\partial \mathcal{G}_0^e$.

**Case I: Node at Last.** If $Q$-value of each state is well estimated, i.e., $Q = Q^*$, then the optimal solution path $\mathcal{P}_{e+1}$ at episode $e+1$ should be close to the optimal solution path $\mathcal{P}_{\text{full}}$ in the full graph $\mathcal{G}_{\text{full}}$, and the last state of the path $\mathcal{P}_{e+1}$ should be closest to the terminal state. Hence, if $g_e^*$ is not in the boundary, there must be one neighbor node closer to the terminal state. Otherwise, $g_e^*$ is the dead end and thus should not be regarded as the optimal goal. And if there is one neighbor node closer to the terminal state, then this state should be regarded as the optimal goal. Therefore, we obtain a contradiction.

**Case II: Node Not at Last.** If the optimal goal is not the last state, then there must exist the state unexplored at episode $e$. Take Figure 10 as an example, if we take $s_6$ as the optimal goal $g_e^*$ in (d), state $s_9$ must be unexplored in $\mathcal{G}_0^e$ in (c) and explored in $\mathcal{G}_0^{e+1}$ in (b). If $g_e^*$ is not included in $\partial \mathcal{G}_0^e$, then there should not exist any unexplored state that is included in its neighborhood. According to the definition of the boundary of the graph, we have proved the proposition by contradiction.

In summary, we have proved the proposition in both two cases by contradiction. $\qquad \square$

### D.3 PROOF OF PROPOSITION 3

**Proposition 3.** *Denote the Bellman backup operator in Eq. (6) as $\mathcal{B} : \mathbb{R}^{|S| \times |A| \times |G|} \rightarrow \mathbb{R}^{|S| \times |A| \times |G|}$ and a mapping $Q : S \times A \times G \rightarrow \mathbb{R}^{|S| \times |A| \times |G|}$ with $|S| < \infty$ and $|A| < \infty$. Repeated applications of the operator $\mathcal{B}$ for our graph-based state-action value estimate $\hat{Q}_\mathcal{G}$ converges to a unique optimal value $\hat{Q}_{\mathcal{G}^*}^*$ with well-explored graph $\mathcal{G}^*$ (i.e., fully explored graph $\mathcal{G}_{\text{full}}$) including optimal solution path.*

*Proof.* The proof of Proposition 3 is done in two main steps. The first step is to show that our state-transition graph $\mathcal{G}$ can converge to the well-explored graph $\mathcal{G}^*$. Here, we define $\mathcal{G}^*$ as the graph that includes the optimal path (i.e., $\mathcal{P}_{\text{full}}$ in Definition 2). In the second step, we prove that given graph $\mathcal{G}$, our graph-based method can converge to unique optimal value $Q_\mathcal{G}^*$.

**Step I.** Since $|S| < \infty$ and $|A| < \infty$, we can obtain that $\mathcal{V}_\mathcal{G} < \infty$ and $\mathcal{E}_\mathcal{G} < \infty$. Note that the state-transition graph $\mathcal{G}$ is a dynamic graph, and goals $g$ generated on $\mathcal{G}$ are updated at the beginning timestep of each episode. Hence, there is a sequence of goals denoted as $(g_1, g_2, \cdots, g_E)$ and corresponding sequence of graphs denoted as $(\mathcal{G}_0^1, \mathcal{G}_0^2, \cdots, \mathcal{G}_0^E)$, where $E$ here is the number of episodes. Given that $|S| < \infty$ and $|A| < \infty$, the number of nodes and edges in the full graph $\mathcal{G}_{\text{full}}$ is also bounded. Based on the explore strategy introduced in Section 3, we know that goal-oriented RL will first search for a path leading to the terminal state. After that, the terminal state will be included in $\mathcal{G}$. Then the agent will seek the shortest path to the terminal state because the agent is given a negative reward at each timestep. Hence, the optimal solution path $\mathcal{P}_{\text{full}}$ will be involved. Hence, we can obtain that

$$\mathcal{G}_0^1 \subseteq \mathcal{G}_0^2 \subseteq \cdots \subseteq \mathcal{G}^* \Rightarrow \mathcal{G} \to \mathcal{G}^*. \tag{13}$$

Assume that $E$ is large enough, our state-transition graph $\mathcal{G}$ can finally converge to well-explored graph $\mathcal{G}^*$.

**Step II.** Note that the proof of convergence for our graph-based goal-oriented RL is quite similar to $Q$-learning (Bellman, 1966; Bertsekas, 1995; Sutton & Barto, 2018). The differences between our approach and $Q$-learning are that $Q$ value $Q(s, a, g)$ is also conditioned on goal $g$, and that the state-transition probability $P_\mathcal{G}(s'|s, a)$ can be reflected by graph $\mathcal{G}$. We provide detailed proof as follows:

For any state-transition graph $\mathcal{G}$, we can obtain goal $g \in G$ conditioned on $\mathcal{G}$ from Step I. Based on that, our estimated graph-based action-value function $\hat{Q}_\mathcal{G}$ can be defined as

$$\mathcal{B}\hat{Q}_\mathcal{G}(s, a, g) = R(s, a, g) + \gamma \cdot \max_{a' \in A} \sum_{s' \in S} P_\mathcal{G}(s'|s, a) \cdot \hat{Q}_\mathcal{G}(s', a', g). \tag{14}$$

For any action-value function estimates $\hat{Q}_\mathcal{G}^1, \hat{Q}_\mathcal{G}^2$, we study that

$$\begin{aligned}
&|\mathcal{B}\hat{Q}_\mathcal{G}^1(s, a, g) - \mathcal{B}\hat{Q}_\mathcal{G}^2(s, a, g)| \\
&= \gamma \cdot |\max_{a' \in A} \sum_{s' \in S} P_\mathcal{G}(s'|s, a) \cdot \hat{Q}_\mathcal{G}^1(s', a', g) - \max_{a' \in A} \sum_{s' \in S} P_\mathcal{G}(s'|s, a) \cdot \hat{Q}_\mathcal{G}^2(s', a', g)| \\
&\leq \gamma \cdot \max_{a' \in A} |\sum_{s' \in S} P_\mathcal{G}(s'|s, a) \cdot \hat{Q}_\mathcal{G}^1(s', a', g) - \sum_{s' \in S} P_\mathcal{G}(s'|s, a) \cdot \hat{Q}_\mathcal{G}^2(s', a', g)| \\
&= \gamma \cdot \max_{a' \in A} \sum_{s' \in S} P_\mathcal{G}(s'|s, a) \cdot |\hat{Q}_\mathcal{G}^1(s', a', g) - \hat{Q}_\mathcal{G}^2(s', a', g)| \\
&\leq \gamma \cdot \max_{s \in S, a \in A} |\hat{Q}_\mathcal{G}^1(s, a, g) - \hat{Q}_\mathcal{G}^2(s, a, g)|
\end{aligned} \tag{15}$$

So the contraction property of Bellman operator holds that

$$\max_{s \in S, a \in A} |\mathcal{B}\hat{Q}_\mathcal{G}^1(s, a, g) - \mathcal{B}\hat{Q}_\mathcal{G}^2(s, a, g)| \leq \gamma \cdot \max_{s \in S, a \in A} |\hat{Q}_\mathcal{G}^1(s, a, g) - \hat{Q}_\mathcal{G}^2(s, a, g)| \tag{16}$$

For the fixed point $Q_\mathcal{G}^*$, we have that

$$\max_{s \in S, a \in A} |\mathcal{B}\hat{Q}_\mathcal{G}(s, a, g) - \mathcal{B}\hat{Q}_\mathcal{G}^*(s, a, g)| \leq \gamma \cdot \max_{s \in S, a \in A} |\hat{Q}_\mathcal{G}(s, a, g) - \hat{Q}_\mathcal{G}^*(s, a, g)| \Rightarrow \hat{Q}_\mathcal{G} \to Q_\mathcal{G}^*. \tag{17}$$

Combining Step I and II, we can conclude that our graph-based estimated state-action value $\hat{Q}_\mathcal{G}$ can converge to a unique optimal value $Q_{\mathcal{G}^*}^*$. $\qquad\square$

# E  DISCUSSIONS

## E.1  DISCUSSION ON CERTAINTY OF STATE

In this section, we further discuss the relationship between the certainty of state and the number of states. In the previous exploration RL literature (Ostrovski et al., 2017; Bellemare et al., 2016), the performance of exploration often is measured by the number of the visited states. Namely, given a fixed number of episodes, more visited states, better performance. In this paper, we propose to utilize a new measurement, *i.e.*, certainty of state as illustrated in Definition 1. We conclude the relations between certainty and the number of visited states as Proposition 4.

Figure 11: An illustrated example for relationship between certainty and number of visited states.

**Proposition 4.** *Given a whole state-transition graph $\mathcal{G}_{whole}$, we can regard the certainty of states as the local measurement and the number of states as the global measurement for exploration, which share a similar trend during agent exploration.*

*Proof.* We illustrate and prove the proposition hindsightly. If we have the full observation for states as shown in Figure 11(a), we can model the agent finding new states as connecting new states with visited states. In other words, since the state-transition graph $\mathcal{G}_t^e$ must keep being a fully connected graph at any timestep $t$ in any episode $e$. Hence, adding new states into the visited state set can always be regarded as finding new edge between new states and the visited state set. And each directed edge in the state-transition graph, as shown in Figure 11(b) is determined by action and state-transition function. If the environment is determined, we can roughly regard the number of edges as the approximate measurement for exploration. The certainty of states is the local perspective for this measurement. □

### E.2 DISCUSSION ON OPTIMAL GOAL

In the previous goal-oriented RL literature (Andrychowicz et al., 2017; Ren et al., 2019), what kind of generated goals is helpful for the agent to efficiently learn a well-performed policy is one of the key questions to be answered. The basic idea of goal-oriented RL architecture is to generate goals to decompose the complex task into several goal-oriented tasks. In this paper, we analyze our generated goals from two perspectives, namely reachability and curriculum.

**Reachability**. The first property required in the optimal goal is that the generated goal is guaranteed to be reachable for the agent. To this end, in this paper, the candidate goal set is constrained into the visited states. In other words, the goal generated in the episode $e$ must be visited before the episode $e$. Therefore, we can guarantee that the generated goal is reachable.

**Curriculum**. The second property is the curriculum, which means that our optimal goals are required to approach the terminal state during the exploration. If the $Q$-value of each state is well estimated, our goal generation under the supervision of forward-looking planning at the next episode will focus on the potential highest value states in the future, which is actually the terminal state when the agent has the full observation of states.

### E.3 DISCUSSION ON GROUP DIVISION

**Motivation**. The intuitive motivation behind the group division is very natural. Proposition 1 implies that exploration on the state-transition graph $\mathcal{G}_t^e$ at timestep $t$ in episode $e$ without any constraint may lead to explosion of graph and inefficiency of exploration. Therefore, the agent is expected to do exploration within a limited domain. Considering that $\mathcal{G}_t^e$ is always changing and the number of nodes (*i.e.*, $|S_{\mathcal{G}_t^e}|$) keeps increasing, it is non-trivial for the agent to learn to select state as the goal for further exploration. Hence, we first restrict the exploration within the boundary of state-transition graph $\partial\mathcal{G}_t^e$ according to Proposition 2. We then consider partitioning $\partial\mathcal{G}_t^e$ into several groups.

We set the last visited state $s_{last}$ as the original point because $s_{last}$ is likely to be close to the target state and reachable for current policy. As introduced in Section 3, we propose to extend groups from $s_{last}$ following two possible perspectives, namely neighbor and uncertain nodes.

**Complexity**. Let $d_{\partial\mathcal{G}_t^e}$ denote the maximum degree of states in $\partial\mathcal{G}_t^e$, and $|S_{\partial\mathcal{G}_t^e}|$ denote the number of states in $\partial\mathcal{G}_t^e$. Note that $\partial\mathcal{G}_t^e$ is always a directed fully connected graph. If we want to find the $n$-hop neighbors of $s_{last}$, we need to iteratively go through related nodes' neighborhood. In other words, the computation complexity should be $\mathcal{O}(d_{\partial\mathcal{G}_t^e}^n)$. Hence, the complexity to construct $\mathcal{C}_1, \ldots, \mathcal{C}_N$ by extending from neighbor nodes is $\mathcal{O}(d_{\partial\mathcal{G}_t^e}^1) + \mathcal{O}(d_{\partial\mathcal{G}_t^e}^2) + \cdots + \mathcal{O}(d_{\partial\mathcal{G}_t^e}^{N-1}) = \mathcal{O}(d_{\partial\mathcal{G}_t^e}^{N-1})$. If we want to find nodes whose uncertainty equals $n$, we need to go through the graph once. In this case,

the computation complexity should be $\mathcal{O}(|S_{\partial \mathcal{G}_t^e}|)$. Hence, the complexity to construct $\mathcal{C}_1, \ldots, \mathcal{C}_N$ extending from uncertain nodes is $\mathcal{O}(|S_{\partial \mathcal{G}_t^e}|)$.

## F  EXPERIMENTS

### F.1  ENVIRONMENT CONFIGURATION

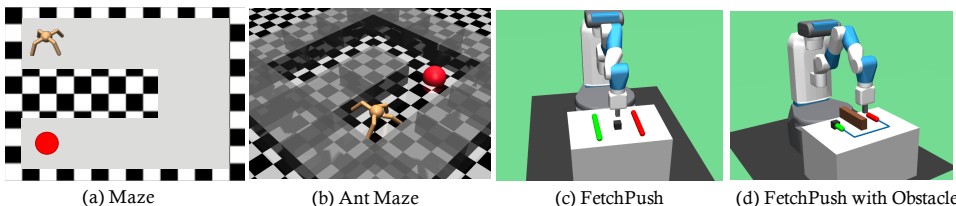

(a) Maze        (b) Ant Maze        (c) FetchPush        (d) FetchPush with Obstacle

Figure 12: Visualization of robotic manipulation environments.

**Maze**. As shown in Figure 12(a), in the maze environment, a point in a $2D$ $U$-maze aims to reach a goal represented by a red point. The size of maze is $15 \times 15$, the state space and is in this $2D$ $U$-maze, and the goal is uniformly generated on the segment from $(0, 0)$ to $(15.0, 15.0)$. The action space is from $(-1.0, -1.0)$ to $(1.0, 1.0)$, which represents the movement in $x$ and $y$ directions.

**AntMaze**. As shown in Figure 12(b), in the AntMaze environment, an ant is put in a $U$-maze, and the size of the maze is $12 \times 12$. The ant is put on a random location on the segment from $(-2.0, -2.0)$ to $(10.0, 10.0)$, and the goal is uniformly generated on the segment from $(-2.0, -2.0)$ to $(10.0, 10.0)$. The state of ant is 30-dimension, including its positions and velocities.

**FetchPush**. As shown in Figure 12(c), in the fetch environment, the agent is trained to fetch an object from the initial position (rectangle depicted in green) to a distant position (rectangle depicted in red). Let the origin $(0, 0, 0)$ denote the projection of the gripper's initial coordinate on the table. The object is uniformly generated on the segment from $(-0.0, -0.0, 0)$ to $(8, 8, 0)$, and the goal is uniformly generated on the segment from $(-0.0, -0.0, 0)$ to $(8, 8, 0)$.

**FetchPush with Obstacle**. As shown in Figure 12(d), in the fetch with obstacle environment, we create an environment based on FetchPush with a rigid obstacle, where the brown block is a static wall that can't be moved. The object is uniformly generated on the segment from $(-0.0, -0.0, 0)$ to $(8, 8, 0)$, and the goal is uniformly generated on the segment from $(-0.0, -0.0, 0)$ to $(8, 8, 0)$.

**AntMaze with Obstacle**. This environment is an extended version of AntMaze, where a $1 \times 1$ rigid obstacle is put in U-maze.

### F.2  EVALUATION DETAILS

- All curves presented in this paper are plotted from 10 runs with random task initialization and seeds.
- The shaded region indicates $60\%$ population around the median.
- All curves are plotted using the same hyper-parameters (except the ablation section).
- Following (Andrychowicz et al., 2017), an episode is considered successful if $\|g - s_{\text{object}}\|_2 \leq \delta_g$ is achieved, where $s_{\text{object}}$ is the object position at the end of the episode. $\delta_g$ is the threshold.
- The max timestep for each episode is set as 200 for training and 500 for tests.
- The average success rate using in the curve is estimated by $10^2$ samples.

### F.3  HYPER-PARAMETERS

Almost all hyper-parameters using DDPG (Lillicrap et al., 2015) and HER Andrychowicz et al. (2017) are kept the same as benchmark results, except these:

- Number of MPI workers: 1;
- Actor and critic networks: 3 layers with 256 units and ReLU activation;
- Adam optimizer with $5 \times 10^{-4}$ learning rate;
- Polyak-averaging coefficient: 0.98;
- Action $l_2$-norm penalty coefficient: 0.5;
- Batch size: 256;

- Probability of random actions: $0.2$;
- Scale of additive Gaussian noise: $0.2$;
- Probability of HER experience replay: $0.8$;
- Number of batches to replay after collecting one trajectory: $50$.

Hyper-parameters in goal generation:

- Adam optimizer with $1 \times 10^{-3}$ learning rate;
- $K$ of $K$-bins discretization: $20$;
- Number of groups to depart the graph: $3$.

## F.4 IMPACT OF GROUP SELECTION

We provide a separate numerical table of asymptotic performance values besides Figure 6(g) here, since the curves of all the models (GSRL and its variants) are close to each other.

| Model | NOGroup | UNCERT2 | UNCERT3 | UNCERT4 | NEIGH3 |
|---|---|---|---|---|---|
| Success Rate | $0.98 \pm 0.03$ | $0.98 \pm 0.02$ | $0.99 \pm 0.02$ | $0.97 \pm 0.03$ | $0.97 \pm 0.03$ |

Table 1: Results of investigation on impact of group selection. Note that the success rate is limited between $0.00$ and $1.00$.

## F.5 COMPARISON ON EXPLORATION

By sample efficiency, we show the comparisons according to the number of states visited and actions taken. In other words, given the fix number of episodes, more unique states visited and actions taken usually denote the efficiency of exploration. We report the log files of GSRL and HER in Maze environment here at $10, 50, 100$ episodes, which contain the number of visited nodes and actions taken.

```
=================== Graph Structured Reinforcement Learning (GSRL) ===================
episode is: 10
nodes: [22, 21, 11, 31, 42, 32, 41, 23, 43, 33, 44, 34, 13, 14, 15, 26, 25, 36, 35, 45, 16, 24, 37, 47, 38, 48, 46, 12, 49,
    59, 69, 79, 80, 78, 90, 89, 99, 109, 110, 100, 27, 28, 39, 50, 40]
number of nodes: 45
edges: [(22, 22), (22, 21), (22, 23), (22, 32), (22, 33), (22, 13), (22, 31), (22, 12), (21, 21), (21, 11), (21, 31), (21,
    32), (21, 22), (11, 11), (11, 21), (11, 12), (31, 31), (31, 42), (31, 41), (31, 22), (31, 32), (31, 21), (42, 42), (42,
    32), (42, 43), (42, 41), (42, 31), (32, 31), (32, 32), (32, 42), (32, 43), (32, 33), (32, 23), (32, 22), (41, 41),
    (41, 31), (23, 22), (23, 23), (23, 32), (23, 33), (23, 24), (43, 32), (43, 43), (43, 42), (43, 33), (43, 44),
    (43, 34), (33, 33), (33, 44), (33, 32), (33, 43), (33, 23), (33, 34), (44, 44), (44, 33), (44, 43), (44, 35), (44, 34)
    , (44, 45), (34, 43), (34, 33), (34, 34), (34, 24), (34, 35), (34, 44), (34, 45), (13, 13), (13, 14), (13, 23), (14,
    15), (14, 25), (14, 14), (15, 15), (15, 26), (15, 25), (15, 14), (15, 16), (26, 26), (26, 25), (26, 36), (26, 16), (26,
    37), (26, 27), (25, 26), (25, 25), (25, 15), (25, 36), (36, 35), (36, 26), (36, 36), (36, 37), (36, 27), (35, 35),
    (35, 45), (35, 26), (35, 34), (35, 36), (35, 44), (35, 25), (45, 35), (45, 44), (45, 45), (16, 26), (24, 35), (24, 34),
    (24, 25), (37, 47), (37, 37), (37, 38), (37, 48), (47, 37), (47, 47), (47, 48), (47, 46), (38, 47), (38, 48), (38, 37)
    , (38, 49), (38, 28), (38, 39), (48, 38), (48, 48), (48, 49), (46, 47), (46, 46), (12, 11), (12, 21), (12, 23), (12,
    22), (12, 13), (49, 48), (49, 59), (49, 50), (59, 69), (69, 69), (69, 79), (79, 80), (79, 78), (79, 79), (79, 90), (80,
    79), (78, 79), (90, 89), (89, 99), (99, 99), (99, 109), (109, 110), (109, 109), (109, 100), (110, 100), (100, 109),
    (27, 36), (27, 27), (27, 38), (28, 28), (28, 38), (39, 50), (39, 40), (39, 39), (50, 39), (50, 40), (50, 50), (40, 49),
    (40, 39), (40, 50)]
number of edges: 166
episode is: 50
nodes: [22, 21, 11, 31, 42, 32, 41, 23, 43, 33, 44, 34, 13, 14, 15, 26, 25, 36, 35, 45, 16, 24, 37, 47, 38, 48, 46, 12, 49,
    59, 69, 79, 80, 78, 90, 89, 99, 109, 110, 100, 27, 28, 39, 50, 40, 29, 30, 51, 57, 18, 19, 58, 68, 17, 60, 20, 67, 70,
    71, 61]
number of nodes: 60
edges: [(22, 22), (22, 21), (22, 23), (22, 32), (22, 33), (22, 13), (22, 31), (22, 12), (22, 11), (21, 21), (21, 11), (21,
    31), (21, 32), (21, 22), (21, 12), (11, 11), (11, 21), (11, 12), (11, 22), (31, 31), (31, 42), (31, 41), (31, 22), (31,
    32), (31, 21), (31, 40), (31, 30), (42, 42), (42, 32), (42, 43), (42, 41), (42, 31), (32, 31), (32, 32), (32, 42),
    (32, 43), (32, 33), (32, 32), (32, 41), (32, 41), (41, 41), (41, 31), (41, 40), (41, 32), (23, 22), (23, 23),
    (23, 32), (23, 34), (23, 33), (23, 24), (23, 13), (23, 14), (23, 12), (43, 32), (43, 43), (43, 42), (43, 33), (43, 44)
    , (43, 34), (33, 33), (33, 44), (33, 32), (33, 43), (33, 23), (33, 34), (33, 22), (33, 42), (33, 24), (44, 44), (44,
    33), (44, 43), (44, 35), (44, 34), (44, 45), (34, 43), (34, 33), (34, 34), (34, 24), (34, 35), (34, 44), (34, 45), (34,
    25), (13, 13), (13, 14), (13, 23), (13, 12), (13, 22), (13, 24), (14, 15), (14, 25), (14, 14), (14, 24), (14, 13),
    (14, 23), (15, 15), (15, 26), (15, 25), (15, 14), (15, 16), (26, 26), (26, 25), (26, 36), (26, 16), (26, 37), (26, 27),
    (26, 35), (26, 17), (25, 26), (25, 25), (25, 15), (25, 36), (25, 24), (25, 35), (25, 16), (25, 34), (36, 35), (36, 26)
    , (36, 36), (36, 37), (36, 27), (36, 46), (36, 47), (36, 45), (35, 35), (35, 45), (35, 26), (35, 34), (35, 36), (35,
    44), (35, 25), (35, 46), (35, 24), (45, 45), (45, 44), (45, 46), (45, 36), (45, 34), (16, 26), (16, 16), (16,
    27), (16, 17), (16, 15), (16, 25), (24, 35), (24, 34), (24, 25), (24, 24), (24, 14), (24, 23), (24, 15), (24, 33),
    (37, 47), (37, 37), (37, 38), (37, 48), (37, 28), (37, 36), (37, 46), (37, 27), (47, 37), (47, 47), (47, 48), (47, 46),
    (47, 38), (47, 58), (47, 57), (47, 36), (38, 47), (38, 48), (38, 37), (38, 49), (38, 28), (38, 39), (38, 38), (38, 27)
    , (48, 38), (48, 48), (48, 49), (48, 57), (48, 47), (48, 58), (48, 59), (46, 47), (46, 46), (46, 37), (46, 45), (46,
```

36), (46, 35), (12, 11), (12, 21), (12, 23), (12, 22), (12, 13), (12, 12), (49, 48), (49, 59), (49, 50), (49, 39), (49, 49), (49, 60), (59, 69), (59, 59), (59, 48), (59, 50), (59, 60), (59, 49), (59, 58), (69, 69), (69, 79), (69, 78), (69, 80), (79, 80), (79, 78), (79, 79), (79, 90), (79, 68), (80, 79), (80, 69), (80, 80), (80, 90), (78, 79), (78, 78), (78, 68), (78, 69), (78, 89), (90, 89), (90, 79), (90, 90), (89, 99), (89, 79), (99, 99), (99, 109), (109, 110), (109, 109), (109, 100), (110, 100), (100, 109), (27, 36), (27, 27), (27, 38), (27, 28), (27, 26), (27, 37), (27, 18), (27, 16), (27, 17), (28, 28), (28, 38), (28, 27), (28, 18), (28, 19), (28, 37), (28, 39), (28, 29), (39, 50), (39, 40), (39, 39), (39, 29), (39, 38), (50, 39), (50, 40), (50, 50), (50, 49), (50, 59), (50, 60), (50, 51), (40, 49), (40, 39), (40, 50), (40, 40), (40, 51), (40, 41), (40, 29), (40, 31), (40, 30), (29, 30), (29, 39), (29, 29), (29, 19), (29, 28), (30, 29), (30, 40), (30, 31), (30, 30), (30, 20), (30, 39), (51, 40), (57, 57), (57, 68), (57, 47), (57, 58), (57, 48), (18, 19), (18, 27), (18, 28), (18, 18), (18, 17), (18, 29), (19, 28), (19, 19), (19, 18), (19, 30), (19, 29), (58, 48), (58, 58), (58, 57), (58, 59), (58, 47), (58, 67), (68, 69), (68, 78), (68, 79), (17, 17), (17, 18), (17, 28), (17, 16), (17, 27), (60, 50), (60, 60), (60, 49), (60, 70), (60, 61), (60, 59), (20, 19), (67, 67), (67, 58), (70, 71), (70, 70), (70, 60), (70, 69), (71, 71), (71, 70), (61, 70)]
number of edges: 336
episode: 100
nodes: [22, 21, 11, 31, 42, 32, 41, 23, 43, 33, 44, 34, 13, 14, 15, 26, 25, 36, 35, 45, 16, 24, 37, 47, 38, 48, 46, 12, 49, 59, 69, 79, 80, 78, 90, 89, 99, 109, 110, 100, 27, 28, 39, 50, 40, 29, 30, 51, 57, 18, 19, 58, 68, 17, 60, 20, 67, 70, 71, 61, 88, 87, 96, 106, 105, 104, 114, 115, 81, 77, 97, 107, 86, 98, 108, 95, 85, 94, 103]
number of nodes: 79
edges: [(22, 22), (22, 21), (22, 23), (22, 32), (22, 33), (22, 13), (22, 31), (22, 12), (22, 11), (21, 21), (21, 11), (21, 31), (21, 32), (21, 22), (21, 12), (21, 20), (21, 30), (11, 11), (11, 21), (11, 12), (11, 22), (31, 31), (31, 42), (31, 41), (31, 22), (31, 32), (31, 21), (31, 40), (31, 30), (42, 42), (42, 32), (42, 43), (42, 41), (42, 31), (42, 33), (32, 31), (32, 32), (32, 42), (32, 43), (32, 33), (32, 22), (32, 41), (32, 21), (41, 41), (41, 31), (41, 40), (41, 32), (41, 42), (41, 50), (41, 30), (23, 22), (23, 23), (23, 32), (23, 34), (23, 33), (23, 24), (23, 13), (23, 14), (23, 12), (43, 32), (43, 43), (43, 42), (43, 33), (43, 44), (43, 34), (33, 33), (33, 44), (33, 32), (33, 43), (33, 23), (33, 34), (33, 22), (33, 42), (33, 24), (44, 44), (44, 33), (44, 43), (44, 35), (44, 34), (44, 45), (34, 43), (34, 33), (34, 34), (34, 24), (34, 35), (34, 44), (34, 25), (34, 23), (13, 13), (13, 14), (13, 23), (13, 12), (13, 22), (13, 24), (14, 15), (14, 25), (14, 14), (14, 24), (14, 13), (14, 23), (15, 15), (15, 26), (15, 25), (15, 14), (15, 16), (15, 24), (26, 26), (26, 25), (26, 36), (26, 16), (26, 37), (26, 27), (26, 35), (26, 17), (26, 15), (25, 26), (25, 25), (25, 15), (25, 36), (25, 24), (25, 35), (25, 16), (25, 34), (25, 14), (36, 35), (36, 26), (36, 36), (36, 37), (36, 27), (36, 46), (36, 47), (36, 45), (36, 25), (35, 35), (35, 45), (35, 26), (35, 34), (35, 36), (35, 44), (35, 25), (35, 46), (35, 24), (45, 35), (45, 44), (45, 45), (45, 46), (45, 36), (45, 34), (16, 26), (16, 16), (16, 27), (16, 17), (16, 15), (16, 25), (24, 35), (24, 34), (24, 25), (24, 24), (24, 14), (24, 23), (24, 15), (24, 33), (24, 13), (37, 47), (37, 37), (37, 38), (37, 48), (37, 28), (37, 36), (37, 46), (37, 27), (37, 26), (47, 37), (47, 47), (47, 48), (47, 46), (47, 38), (47, 58), (47, 57), (47, 36), (38, 47), (38, 48), (38, 37), (38, 49), (38, 28), (38, 39), (38, 38), (38, 27), (38, 29), (48, 38), (48, 48), (48, 49), (48, 57), (48, 47), (48, 58), (48, 59), (48, 37), (48, 39), (46, 47), (46, 46), (46, 37), (46, 45), (46, 36), (46, 35), (12, 11), (12, 21), (12, 23), (12, 22), (12, 13), (12, 12), (49, 48), (49, 59), (49, 50), (49, 39), (49, 49), (49, 60), (49, 58), (49, 40), (49, 38), (59, 69), (59, 59), (59, 48), (59, 50), (59, 60), (59, 49), (59, 58), (59, 68), (59, 70), (69, 69), (69, 79), (69, 78), (69, 80), (69, 70), (69, 68), (69, 59), (79, 80), (79, 78), (79, 79), (79, 90), (79, 68), (79, 88), (79, 89), (79, 69), (80, 79), (80, 69), (80, 80), (80, 90), (80, 89), (80, 81), (80, 70), (80, 71), (78, 79), (78, 78), (78, 68), (78, 69), (78, 89), (78, 87), (78, 67), (78, 77), (78, 88), (90, 89), (90, 79), (90, 90), (90, 80), (89, 99), (89, 79), (89, 80), (89, 89), (89, 88), (89, 90), (89, 78), (89, 98), (99, 99), (99, 109), (99, 88), (99, 89), (99, 98), (99, 100), (109, 110), (109, 109), (109, 100), (110, 100), (100, 109), (100, 99), (27, 36), (27, 27), (27, 38), (27, 28), (27, 26), (27, 37), (27, 18), (27, 16), (27, 17), (28, 28), (28, 38), (28, 27), (28, 18), (28, 19), (28, 37), (28, 39), (28, 29), (28, 17), (39, 50), (39, 40), (39, 39), (39, 29), (39, 38), (39, 49), (39, 48), (39, 30), (50, 39), (50, 40), (50, 50), (50, 49), (50, 59), (50, 60), (50, 51), (50, 61), (50, 41), (40, 49), (40, 39), (40, 50), (40, 40), (40, 51), (40, 41), (40, 29), (40, 31), (40, 30), (29, 30), (29, 39), (29, 29), (29, 19), (29, 28), (29, 18), (29, 40), (29, 38), (30, 29), (30, 40), (30, 31), (30, 30), (30, 20), (30, 39), (30, 19), (30, 21), (51, 40), (51, 51), (51, 60), (51, 50), (57, 57), (57, 68), (57, 47), (57, 58), (57, 48), (57, 67), (18, 19), (18, 27), (18, 28), (18, 18), (18, 17), (18, 29), (19, 28), (19, 19), (19, 18), (19, 30), (19, 29), (19, 20), (58, 48), (58, 58), (58, 57), (58, 59), (58, 49), (58, 47), (58, 67), (58, 69), (58, 68), (68, 69), (68, 78), (68, 79), (68, 68), (68, 57), (68, 67), (68, 77), (68, 58), (17, 17), (17, 18), (17, 28), (17, 16), (17, 27), (60, 50), (60, 60), (60, 49), (60, 70), (60, 61), (60, 59), (60, 51), (60, 69), (60, 71), (20, 19), (20, 30), (20, 20), (20, 21), (20, 29), (67, 67), (67, 58), (67, 68), (67, 78), (67, 77), (67, 57), (70, 71), (70, 70), (70, 60), (70, 69), (70, 79), (70, 80), (70, 59), (71, 71), (71, 70), (71, 80), (61, 70), (61, 61), (61, 50), (61, 60), (88, 87), (88, 79), (88, 89), (88, 88), (88, 99), (88, 78), (88, 97), (87, 78), (87, 88), (87, 96), (87, 87), (87, 97), (87, 77), (87, 86), (96, 106), (96, 97), (96, 87), (96, 86), (96, 96), (96, 95), (106, 105), (106, 107), (106, 96), (105, 105), (105, 104), (105, 114), (105, 115), (104, 114), (104, 104), (104, 105), (114, 114), (114, 104), (114, 105), (115, 105), (81, 80), (77, 77), (77, 67), (77, 68), (77, 88), (77, 78), (97, 96), (97, 106), (97, 107), (97, 87), (107, 96), (107, 106), (107, 107), (107, 108), (86, 96), (86, 86), (98, 99), (98, 89), (98, 98), (95, 95), (95, 85), (95, 94), (85, 85), (85, 95), (94, 103), (103, 103)]
number of edges: 486
==================== Hindsight Experience Replay (HER) ====================
episode is: 10
nodes: [22, 21, 31, 41, 42, 32, 23, 43, 33, 44, 34, 35, 45, 46, 36, 37, 47, 13, 12, 14, 15, 16, 17, 26, 25, 24, 48, 11, 27]
number of nodes: 29
edges: [(22, 21), (22, 23), (22, 22), (22, 32), (22, 33), (22, 12), (21, 21), (21, 31), (21, 22), (31, 31), (31, 41), (31, 21), (31, 42), (41, 42), (41, 41), (41, 32), (41, 31), (42, 41), (42, 42), (42, 32), (42, 43), (42, 31), (32, 31), (32, 32), (32, 42), (32, 33), (32, 23), (32, 41), (23, 22), (23, 13), (23, 14), (23, 33), (43, 43), (43, 42), (43, 33), (43, 44), (43, 34), (33, 33), (33, 44), (33, 32), (33, 43), (33, 34), (44, 44), (44, 33), (44, 43), (44, 34), (44, 35), (44, 45), (34, 43), (34, 33), (34, 35), (34, 34), (34, 45), (35, 45), (35, 46), (35, 35), (35, 34), (35, 25), (45, 46), (45, 45), (45, 35), (45, 44), (45, 34), (46, 46), (46, 45), (46, 35), (46, 36), (46, 37), (36, 36), (36, 37), (36, 47), (36, 26), (36, 46), (37, 47), (37, 37), (37, 36), (37, 48), (47, 47), (47, 37), (47, 46), (13, 13), (13, 12), (13, 14), (12, 13), (12, 11), (12, 12), (12, 23), (14, 14), (14, 13), (14, 15), (15, 15), (15, 16), (15, 26), (15, 25), (16, 16), (16, 17), (26, 25), (26, 26), (26, 37), (26, 36), (26, 27), (25, 24), (25, 36), (25, 26), (24, 15), (48, 37), (11, 12), (11, 11), (27, 27)]
number of edges: 112
episode is: 50
nodes: [22, 21, 31, 41, 42, 32, 23, 43, 33, 44, 34, 35, 45, 46, 36, 37, 47, 13, 12, 14, 15, 16, 17, 26, 25, 24, 48, 11, 27, 18, 19, 20, 58, 57, 49, 39, 60, 50, 59, 40, 38, 28, 29, 30, 69, 70, 80, 90, 101, 100, 67, 68, 77, 78, 61]
number of nodes: 55

edges: [(22, 21), (22, 23), (22, 22), (22, 32), (22, 33), (22, 12), (22, 13), (22, 11), (21, 21), (21, 31), (21, 22), (21, 11), (21, 12), (21, 32), (31, 31), (31, 41), (31, 21), (31, 42), (31, 32), (41, 42), (41, 41), (41, 32), (41, 31), (42, 41), (42, 42), (42, 32), (42, 43), (42, 31), (42, 33), (32, 23), (32, 41), (32, 22), (32, 21), (23, 22), (23, 13), (23, 14), (23, 33), (23, 34), (23, 24), (23, 23), (23, 12), (43, 32), (43, 43), (43, 42), (43, 33), (43, 44), (43, 34), (33, 33), (33, 44), (33, 32), (33, 43), (33, 34), (33, 23), (33, 22), (33, 42), (33, 24), (44, 44), (44, 33), (44, 43), (44, 34), (44, 35), (44, 45), (34, 43), (34, 33), (34, 35), (34, 34), (34, 45), (34, 23), (34, 44), (34, 25), (34, 24), (35, 45), (35, 46), (35, 35), (35, 34), (35, 25), (35, 44), (35, 36), (45, 46), (45, 45), (45, 35), (45, 44), (45, 34), (45, 36), (46, 46), (46, 45), (46, 35), (46, 36), (46, 37), (46, 47), (36, 36), (36, 37), (36, 47), (36, 26), (36, 46), (36, 45), (36, 35), (36, 27), (37, 47), (37, 37), (37, 36), (37, 48), (37, 46), (37, 27), (47, 47), (47, 37), (47, 36), (47, 46), (47, 48), (47, 38), (47, 57), (47, 58), (13, 13), (13, 14), (13, 12), (13, 22), (13, 23), (12, 13), (12, 11), (12, 12), (12, 23), (12, 22), (12, 21), (14, 14), (14, 13), (14, 15), (14, 23), (14, 25), (14, 24), (15, 15), (15, 16), (15, 26), (15, 25), (15, 14), (16, 16), (16, 17), (16, 15), (16, 26), (17, 17), (17, 18), (26, 25), (26, 26), (26, 37), (26, 36), (26, 27), (26, 15), (26, 16), (25, 24), (25, 36), (25, 26), (25, 16), (25, 15), (25, 14), (25, 25), (25, 35), (24, 15), (24, 35), (24, 24), (24, 25), (24, 33), (24, 23), (24, 34), (48, 37), (48, 47), (48, 48), (48, 59), (48, 49), (48, 59), (48, 38), (11, 12), (11, 11), (11, 21), (27, 27), (27, 28), (27, 38), (27, 37), (18, 18), (18, 19), (19, 19), (19, 20), (20, 20), (20, 19), (20, 30), (58, 57), (58, 69), (58, 58), (58, 67), (58, 59), (58, 48), (57, 48), (57, 57), (57, 58), (57, 67), (49, 39), (49, 60), (49, 50), (49, 59), (49, 49), (39, 49), (39, 29), (39, 39), (60, 49), (60, 50), (60, 70), (50, 60), (50, 49), (50, 40), (59, 49), (59, 59), (59, 69), (59, 58), (59, 60), (40, 49), (38, 47), (38, 28), (38, 38), (38, 39), (38, 48), (28, 29), (28, 38), (29, 29), (29, 30), (29, 39), (30, 30), (30, 20), (30, 29), (69, 69), (69, 70), (69, 58), (69, 68), (69, 59), (70, 70), (70, 80), (70, 61), (80, 80), (80, 90), (90, 90), (90, 101), (101, 101), (101, 100), (100, 101), (100, 100), (100, 90), (67, 68), (67, 67), (67, 77), (68, 58), (68, 69), (77, 77), (77, 78)]
number of edges: 255
episode: 100
nodes: [22, 21, 31, 41, 42, 32, 23, 43, 33, 44, 34, 35, 45, 46, 36, 37, 47, 13, 12, 14, 15, 16, 17, 26, 25, 24, 48, 11, 27, 18, 19, 20, 58, 57, 49, 39, 60, 50, 59, 40, 38, 28, 29, 30, 69, 70, 80, 90, 101, 100, 67, 68, 77, 78, 61, 88, 87, 97, 96, 106, 117, 107, 71, 79, 89, 86, 85, 95, 51, 99, 110]
number of nodes: 71
edges: [(22, 21), (22, 23), (22, 22), (22, 32), (22, 33), (22, 12), (22, 13), (22, 11), (22, 31), (21, 21), (21, 31), (21, 22), (21, 11), (21, 12), (21, 32), (21, 20), (31, 31), (31, 41), (31, 21), (31, 42), (31, 32), (31, 30), (31, 40), (31, 22), (41, 42), (41, 41), (41, 32), (41, 31), (42, 41), (42, 42), (42, 32), (42, 43), (42, 31), (42, 33), (32, 31), (32, 32), (32, 42), (32, 33), (32, 43), (32, 23), (32, 41), (32, 22), (32, 21), (23, 22), (23, 13), (23, 14), (23, 33), (23, 34), (23, 24), (23, 23), (23, 12), (43, 32), (43, 43), (43, 42), (43, 33), (43, 44), (43, 34), (33, 33), (33, 44), (33, 32), (33, 43), (33, 34), (33, 23), (33, 22), (33, 42), (33, 24), (44, 44), (44, 33), (44, 43), (44, 34), (44, 35), (44, 45), (34, 43), (34, 33), (34, 35), (34, 34), (34, 45), (34, 23), (34, 44), (34, 25), (34, 24), (35, 45), (35, 46), (35, 35), (35, 34), (35, 25), (35, 44), (35, 36), (35, 24), (35, 26), (45, 46), (45, 45), (45, 35), (45, 44), (45, 34), (45, 36), (46, 46), (46, 45), (46, 35), (46, 36), (46, 37), (46, 47), (36, 36), (36, 37), (36, 47), (36, 26), (36, 46), (36, 45), (36, 35), (36, 27), (36, 25), (37, 47), (37, 37), (37, 36), (37, 48), (37, 46), (37, 27), (37, 38), (37, 26), (47, 47), (47, 37), (47, 36), (47, 46), (47, 48), (47, 38), (47, 57), (47, 58), (13, 13), (13, 12), (13, 14), (13, 23), (13, 24), (12, 13), (12, 11), (12, 12), (12, 23), (12, 22), (12, 21), (14, 14), (14, 13), (14, 15), (14, 23), (14, 25), (14, 24), (15, 15), (15, 16), (15, 26), (15, 25), (15, 14), (15, 24), (16, 16), (16, 17), (16, 15), (16, 26), (16, 27), (17, 17), (17, 18), (17, 16), (17, 28), (17, 27), (17, 26), (26, 25), (26, 26), (26, 37), (26, 36), (26, 27), (26, 15), (26, 16), (26, 35), (26, 17), (25, 24), (25, 36), (25, 26), (25, 16), (25, 15), (25, 14), (25, 25), (25, 35), (24, 15), (24, 35), (24, 24), (24, 25), (24, 33), (24, 23), (24, 34), (24, 14), (48, 37), (48, 47), (48, 48), (48, 58), (48, 49), (48, 59), (48, 38), (48, 39), (48, 57), (11, 12), (11, 11), (11, 21), (27, 27), (27, 28), (27, 38), (27, 37), (27, 18), (27, 26), (27, 17), (18, 18), (18, 19), (18, 17), (19, 19), (19, 20), (19, 29), (19, 18), (19, 28), (20, 20), (20, 19), (20, 30), (20, 29), (20, 21), (58, 57), (58, 69), (58, 58), (58, 67), (58, 59), (58, 48), (58, 68), (58, 49), (57, 48), (57, 57), (57, 58), (57, 67), (57, 68), (49, 39), (49, 60), (49, 50), (49, 59), (49, 49), (49, 58), (49, 48), (39, 49), (39, 29), (39, 39), (39, 38), (39, 30), (39, 40), (60, 49), (60, 50), (60, 70), (60, 60), (60, 69), (60, 59), (60, 61), (50, 60), (50, 49), (50, 40), (50, 61), (50, 50), (50, 51), (59, 49), (59, 59), (59, 69), (59, 58), (59, 60), (59, 68), (40, 49), (40, 40), (40, 39), (40, 29), (40, 30), (40, 31), (40, 50), (38, 47), (38, 28), (38, 38), (38, 39), (38, 48), (38, 37), (38, 29), (38, 49), (28, 29), (28, 38), (28, 28), (28, 39), (28, 18), (28, 19), (29, 29), (29, 30), (29, 39), (29, 19), (29, 28), (29, 38), (29, 40), (30, 30), (30, 20), (30, 29), (30, 31), (30, 40), (69, 69), (69, 70), (69, 58), (69, 68), (69, 59), (69, 78), (69, 80), (69, 79), (70, 70), (70, 80), (70, 61), (70, 71), (70, 60), (80, 80), (80, 90), (80, 79), (80, 70), (80, 69), (80, 89), (90, 90), (90, 101), (90, 79), (90, 89), (101, 101), (101, 100), (100, 101), (100, 100), (100, 90), (100, 110), (67, 68), (67, 67), (67, 77), (67, 57), (68, 58), (68, 69), (68, 68), (68, 67), (68, 78), (77, 77), (77, 78), (77, 67), (77, 87), (78, 88), (78, 77), (78, 68), (78, 69), (61, 61), (61, 60), (61, 50), (61, 70), (88, 87), (87, 97), (87, 86), (97, 96), (96, 106), (106, 117), (106, 107), (117, 106), (107, 107), (71, 71), (71, 70), (79, 79), (79, 68), (79, 80), (79, 69), (79, 78), (89, 90), (89, 99), (86, 85), (85, 95), (51, 60), (99, 99), (99, 100)]
number of edges: 370

