# OpenReview forum: "Explore with Dynamic Map: Graph Structured Reinforcement Learning"
_ICLR.cc/2021/Conference — Reject_

### Official Review · AnonReviewer3 · 2020-10-22
**Potentially interesting but major issues with clarity**

**Rating:** 4
**Confidence:** 4

**Review:**

This work presents a strategy for improving exploration and efficiency of RL by leveraging the graph structure of an episodic experience buffer. This strategy combines goal-oriented RL with structured exploration. The authors compare their proposed technique to two popular benchmarks for goal-reaching tasks. In addition, the authors provide some theoretical justification for their algorithmic choices.

### Clarity
The biggest issue with this paper is clarity. Grammar is a minor part of this issue, but the major clarity issue is that it's simply quite hard to understand what exactly the authors are proposing/doing in this work. It's very unfortunate, because it seems like there might be some interesting ideas here, but it is very hard to say because so many details are not clear. The best resource for understanding what the authors' proposed technique is the algorithm box and accompanying text in Appendix B, but this is still not enough.
The lack of clarity is perhaps worst, and most crucial, when it comes to describing how the graph is constructed. How do you go from a set of trajectories to a useful graph representation? In addition, it is difficult to form an intuition around the logic of the algorithm -- for instance, how is the generated goal related to the task goal; or, does the multi-episode graph assume that the task goal is the same across episodes? The result of this confusion is that I would **not** be able to implement their algorithm based on the understanding I gain from the paper.
To the authors' credit, they do attempt to provide some high-level descriptions and include several high-quality illustrations of their proposed strategy, but the lack of clarity remains a decisive problem. It's entirely possible that fixing the clarity issues would result in a strong paper, but, unfortunately, it does not currently seem suitable for publication.


### Quality and Originality
This work fits into a recent trend of work using graph-based representations of the environment for structured exploration and planning. In particular, this work is focused more on graph-based exploration. Overall, the paper's clarity issues prevent a detailed assessment of its broader quality, but the high-level idea seems reasonably original. Certainly, this is not the first paper to recognize that trajectories of experience can reveal the environment's structure and that this structure is conveniently represented as a graph. But I am not aware of another paper that leverages this insight in order to learn to generate a curriculum of goals for good exploration. In addition, the author's technique seems agnostic to the composition of the environment, avoiding some shortcomings of prior work.


### Significance
Again, clarity issues make it difficult to assess the significance of the work. This should be expected to reduce the significance, though, since I doubt most readers will get a good sense of how to actually implement the paper's core idea. Since the work does not present many broader insights about RL, learning from goals, or structured exploration, its significance is linked to its practical value, which is currently limited by clarity issues.


**Pros**
- New algorithm for improving exploration in goal-oriented RL
- Experimental validation in complex, continuous domains

**Cons**
- Very difficult to understand the details
- Comparison to prior work is mostly unclear
- Would be difficult to implement based on current description

### Suggestions for improvement
If the authors can present a clear plan for how they will improve the clarity of the paper, I will be happy to consider raising my score. I recommend that the authors make use of more concrete examples. This would help to make the great illustrations (like Figures 2 and 7) more useful. Also, it may help to put the Related Work at the end of the paper. The comparison to previous work might be clearer if it is presented after the proposed technique is fully described.

---

> ### Author Response · Authors · 2020-11-20
> **Response to Review #3**
>
> Thank you for your feedback and suggested experiments. Please also see the main comment above. We hope you will consider increasing your score after seeing our responses.
>
> >> The best resource for understanding what the authors' proposed technique is the algorithm box and accompanying text in Appendix B, but this is still not enough. The lack of clarity is perhaps worst, and most crucial, when it comes to describing how the graph is constructed.
>
> We have provided an illustrated example in Appendix C. In Figure 8 (in Appendix C), we describe how to discretize the environment with continuous state space in 8(a), and answer how to construct a graph from the replay buffer in 8(b). Then, we show how to group the boundary of the graph, and select appropriate in (c). Please see Appendix C for details.
>
> >> How do you go from a set of trajectories to a useful graph representation?
>
> Firstly, we do not learn a graph representation in this paper. We only leverage structure information (e.g., degree, neighborhood) for goal generation and value estimation as illustrated in Figure 1. Secondly, we provide an illustrated example in Appendix C, where we show how to construct a graph from a set of trajectories.
>
> >> In addition, it is difficult to form an intuition around the logic of the algorithm -- for instance, how is the generated goal related to the task goal; or, does the multi-episode graph assume that the task goal is the same across episodes?
>
> Yes, the task goal is the same across episodes. In goal-oriented RL [1] [2] ad Hierarchical RL [3][4], it is very common to generate several goals (i.e., generated goal) to decompose the task (i.e., task goal) into several sub-tasks. Thus, the difference between the generated goal and the task goal is that the task goal (a.k.a, the terminal state) is given from the environment while the generated goals are generated by the designed model. In other words, the task goal will not change unless the task is finished, while the generated goal will change during each generation period.
>
> >> I recommend that the authors make use of more concrete examples. This would help to make the great illustrations (like Figures 2 and 7) more useful.
>
> Sure. We have provided an illustrated example in Appendix C. We also provide illustrations for notations in the paper in Appendix A. Due to the page limitation, we put them in the appendix in order to avoid too significant changes in the main text.
>
> >> Also, it may help to put the Related Work at the end of the paper. The comparison to previous work might be clearer if it is presented after the proposed technique is fully described.
>
> Thanks for your suggestion.  We have moved the related work section at the end of the paper in our updated draft.
>
> [1] Benjamin Eysenbach, et al. Search on the Replay Buffer: Bridging Planning and Reinforcement Learning. NeurIPS, 2019.
>
> [2] Marcin Andrychowicz, et al. Hindsight Experience Replay. NeurIPS, 2017.
>
> [3] Ofir Nachum, et al. Near-Optimal Representation Learning for Hierarchical Reinforcement Learning. ICLR, 2019.
>
> [4] Alexander Sasha Vezhnevets, et al. FeUdal Networks for Hierarchical Reinforcement Learning. ICML, 2017.

---

### Official Review · AnonReviewer2 · 2020-10-28
**Idea is interesting. Some parts are unclear. Experiments are not enough.**

**Rating:** 5
**Confidence:** 3

**Review:**

This paper proposes a new framework, GSRL, to handle the sparse reward challenge and better leverage past experiences. Specifically, it formulates trajectories as a dynamic graph, and generates hindsight-like goals based on sub-group division and attention mechanism. The authors provide theoretical analysis to show the efficiency and converge property of their method. The experimental result shows the proposed method significantly outperforms the baselines.

Strengths:
+ The general idea is reasonable.
+ Appropriate theoretical analysis is attached.
+ The experimental result is good.

Weaknesses:
+ Is it true that the boundary of a graph consists of all observed states, as defined in Section 4.1? The example in Fig. 4 is misunderstanding.
+ It is unclear about how the groups are represented during computing the attention values. Are they embedded as representations (e.g., averaging all state representations within one group) or just identified as one-hot group ids?
+ Can the paper visualize the groups, their assigned attention values, and the selected goal? If a group is selected, is it true that the goal is sampled randomly from this group, or with some priors. If the boundary of a graph consists of too many states, I wonder whether the groups divided from it are still too large, as there are only three groups.
+ It is better to clarify which episode the s_{last} comes from. According to Fig. 4, the last visited state comes from the "next" episode. However, according to the "Attention Strategy" in Section 4.1, s_{last} is appended to each group since the beginning of each episode. So is s_{last} here not from the "future" episode? Please correct me if I misunderstood your illustration.
+ Regarding Appendix E.4, is a method regarded as more sample efficient if it can explore more unique nodes and form a larger group given a fixed number of episodes?
+ Not sure experiments are enough. For example, just comparing to HER. There are some recent works [1, 2] addressing similar problems and tasks. It is better to compare to them or at least mention the difference.

[1] Curriculum-guided Hindsight Experience Replay, NeurIPS-2019

[2] Maximum Entropy Gain Exploration for Long Horizon Multi-goal Reinforcement Learning, ICML-2020

---

> ### Author Response · Authors · 2020-11-20
> **Response to Review #2**
>
> Thank you for your feedback and suggested experiments. Please also see the main comment above. We hope you will consider increasing your score after seeing our responses.
>
> >> Is it true that the boundary of a graph consists of all observed states, as defined in Section 4.1? The example in Fig. 4 is misunderstanding.
>
> No, the boundary of a graph only consists of observed states that are not well explored. In other words, if there is an action in a state that has not been taken yet, then the state should be included in the boundary. Therefore, the example in Fig.4 is not a misunderstanding. We also show an illustration in Appendix A.
>
> >> It is unclear about how the groups are represented during computing the attention values. Are they embedded as representations (e.g., averaging all state representations within one group) or just identified as one-hot group ids?
>
> We have updated the description of the self-attention mechanism with more details in Section 3.1 Attention Strategy. For each group, we have a unique group id, and we build an embedding vector to represent the feature of the group. For example, the group of nodes with low degrees may denote the group of nodes that are not well explored. We update the embedding following Eq. 5.
>
> >> Can the paper visualize the groups, their assigned attention values, and the selected goal? If a group is selected, is it true that the goal is sampled randomly from this group, or with some priors. If the boundary of a graph consists of too many states, I wonder whether the groups divided from it are still too large, as there are only three groups.
>
> It is hard to visualize all the nodes and edges in the experiment since the number of nodes and edges are too large. Instead, we illustrate an example in Appendix C to show how our model works. If a group is selected, the state with the highest value will be selected as the goal as shown in Eq. 4. If the boundary of a graph consists of too many states, it’s true that the number of states in each group is still very large. First, in our setting, the number of groups is hyper-parameter, we investigate the impact of this hyper-parameter in Figure 6(g). The results indicate that in the Maze environment, 3 groups can obtain the best performance. When an environment is more complex, it obtains more nodes, hence the number of groups also can increase. Second, the embedding of each group is learnable (see Section 3.1 Attention Strategy). When we set the number of groups with a larger number,  it may need more data points for learning. Third, we also involve GSRL without group selection (called NOGroup in Figure 6(g)) as a baseline, which indicates that the group selection can truly improve the performance.
>
> >> It is better to clarify which episode the s_{last} comes from. According to Fig. 4, the last visited state comes from the "next" episode. However, according to the "Attention Strategy" in Section 4.1, s_{last} is appended to each group since the beginning of each episode. So is s_{last} here not from the "future" episode? Please correct me if I misunderstood your illustration.
>
> s_{last} here denotes the last state in the previous episodes. We correct it in the draft. Thanks for pointing it out. We also provide an illustrated example in Appendix C where s_7 is s_{last}.
>
> >> Regarding Appendix E.4, is a method regarded as more sample efficient if it can explore more unique nodes and form a larger group given a fixed number of episodes?
>
> Yes. Results in Appendix F.4 (E.4 in the previous version) shows that GSRL can obtain more unique nodes given a fixed number of episodes. This directly implies that GRSL does a better job on exploration than the baseline. In this way, we can obtain more useful data, which usually means more sample efficiency and better results. We revised the draft to clarify this.
>
> >> Not sure experiments are enough. For example, just comparing to HER. There are some recent works [1, 2] addressing similar problems and tasks. It is better to compare to them or at least mention the difference.
>
> Thanks for your suggestion. We compare GSRL with [1] (called CHER in Figure 6) in the experiment section and discuss the relations between GSRL and [1][2] in the related work section. Please see Figure 6 in the experiment section.
>
> [1] Curriculum-guided Hindsight Experience Replay, NeurIPS-2019
>
> [2] Maximum Entropy Gain Exploration for Long Horizon Multi-goal Reinforcement Learning, ICML-202

---

### Official Review · AnonReviewer1 · 2020-10-29
**Official Blind Review #1**

**Rating:** 6
**Confidence:** 3

**Review:**

This paper introduces Graph Structured Reinforcement Learning (GSRL) framework, able to balance exploration and exploitation in RL. Actually, GSRL builds a dynamic graph based on historical trajectories. Then in order to learn from sparse or delayed rewards and  be able to reach a distant goal, it decomposes the main task into a sequence of easier and shorter tasks. An attention strategy has also been proposed that is able to select an appropriate goal for each one of the easiest tasks. Experiments have been conducted on various robotics manipulation tasks showing that GSRL performs better compared to HER and MAP algorithms.

The idea of constructing a dynamic graph on top of the state-space is really interesting. In this case, the agent is able to explore efficiently all the state-space as the graph is expanded at each time-step. Despite the possible merits of this approach on  various robotics manipulation tasks, the applicability of GSRL on other sensitive tasks is not guaranteed. For instance, could GSRL be applied to the task of helicopter controlling where  the safety of the agent is critical? Another point that should be clarified by the authors is the modelling of the self-attention function. It is not clear how does it defined (Eq. 2)? It is not also obvious  for the reader how do you update the attention mechanism (see Eq. 4). I recommend to the authors to give more details about the definition and update of attention strategy and not treat it as blackbox. Moreover, at Eq. 3 the state with the highest value among the states of C_ATT is selected as goal. How do you learn the value function that is not goal-oriented as the Q-function (learned by minimising the objective function of Eq. 6)? Another point that should be discussed is if there is any connection between GSRL and Hierarchical RL. Finally, despite the fact that the paper is well-written, the notation is cluttered in some cases. For instance, $E$ is used both for the episode length and the number of episodes.

---

> ### Author Response · Authors · 2020-11-20
> **Response to Review #1**
>
> Thank you for your feedback and suggested experiments. Please also see the main comment above. We hope you will consider increasing your score after seeing our responses.
>
> >> The applicability of GSRL on other sensitive tasks is not guaranteed. For instance, could GSRL be applied to the task of helicopter controlling where the safety of the agent is critical?
>
> We have evaluated the performance of GSRL on complex and standard benchmark environments ( refer to Section 4). Results show the efficacy of GSRL and confirm that GSRL performs well on those tasks. It would be interesting to see how GSRL will work on tasks like helicopter controlling and study safety in RL; however, those are out of the scope of this paper and we leave it for future works.
>
> >> Another point that should be clarified by the authors is the modeling of the self-attention function. It is not clear how does it defined (Eq. 2)? It is not also obvious for the reader how do you update the attention mechanism (see Eq. 4).
>
> We have provided a detailed description of the attention mechanism in Eq.2 (see Section 3.1 Attention Strategy). For the update part, we used a standard self-supervised learning algorithm to update attention mechanisms under the supervision of group selection in Eq. 4. The detailed information is updated in Section 3.2  Goal Learning Strategy.
>
> >> At Eq. 3 the state with the highest value among the states of C_ATT is selected as goal. How do you learn the value function that is not goal-oriented as the Q-function (learned by minimising the objective function of Eq. 6)?
>
> We have updated the formulation of the value function in Eq.4 (from V(s)->V(s, g)). Then, the value function is goal-oriented as the Q-function (see  Eq. 1). In Eq. 5, only the attention mechanism is updated and the Q-function (along with the value function) is updated according to Eq. 6.
>
> >> Another point that should be discussed is if there is any connection between GSRL and Hierarchical RL.
>
> We discuss the relations between GSRL and Hierarchical RL in the related work section. The common thing between GSRL and HRL is to learn a “manager module” to assign goals to a “worker module”. The key difference is that GSRL leverages graph structure information to generate appropriate goals.
>
> >> the notation is cluttered in some cases. For instance, E is used both for the episode length and the number of episodes.
>
> Thanks. We have fixed it in the revision.

---

### Official Review · AnonReviewer4 · 2020-10-30
**Promising direction, needs more comparison with existing literature and careful ablations**

**Rating:** 6
**Confidence:** 4

**Review:**


**Summary**
This paper proposes graph-structured reinforcement learning (GSRL), which consists of two key components: (1) goal generation, to choose what goals a goal-conditioned agent should follow during an exploration episode, and (2) value estimation, to prioritize experience from highly related trajectories according to local graph structure during value/policy updates.

For (1), the algorithm maintains a “state-transition graph”, essentially a graph of the observed state transitions in the MDP. At a high level, the goal generation should pick goals in the “boundary” of the current state-transition graph for exploration. This “optimal goal” that should be generated can be trained in hindsight by looking at the best trajectory in the next episode (where the best trajectory terminates at a state that is estimated to be the closest to the goal) and finding in it the reachable state from the current episode. During inference, an attention mechanism is used to identify this optimal goal in the current episode.

For (2), GSRL selects trajectories that have states that are in the immediate neighborhood of the current state to train on, rather than sampling transitions uniformly.

Experiments were conducted in several (discretized) robotics environments ranging from maze navigation to robotic arm block manipulation, demonstrating improvement to HER and MAP baseline.

**Positives**:
- I like the idea of using hindsight to assign the target for choosing behavior goals for exploration. Usually, as in HER, hindsight is applied only for the optimization phase to update the value/policy by augmenting the experience data. This approach can directly affect the data collection process itself.
- The paper presents several theoretical results: (1) to motivate the need for directed exploration, (2) showing that the exploration goals should come from the boundary.

**Negatives**:
- There are several works that investigate letting the agent choose its own goals during exploration: GoalGAN [1], DDL [2], MEGA [3], Skew-Fit [4]. Rather than constructing a graph to determine the boundaries/frontier, they use either a discriminator or a density estimator. It would be worth discussing these works in relation to GSRL, and perhaps even including some of them as baselines.
- For prioritizing experience during value estimation, there is also MEP which tries to group trajectories that achieve diverse goals during the update. This should be compared to $\mathcal{D}_{related}$ in line 13 in Algorithm 1.
- There are several ablations that can be included to further gain intuition about GSRL (see questions below)
- Some parts of the paper were confusing to read. In Section 3, the goal-oriented RL framework was introduced, with both discrete and continuous action space. I was confused then later in section 4 as the dynamic graph construction appears to rely on discrete actions in order to have a notion of degree of node $s$.

**Recommendation**:

Overall, I vote for marginally below the acceptance threshold. The motivation and proposed ideas are promising, but I have several concerns about relevant related literature and ablation experiments which would make the paper much stronger and convincing.

**Question**:
1. Ablations:
  1. How does the method perform without using the attention strategy on the groups to pick $C_{ATT}$ first, then obtain the optimal goal within $C_{ATT}$? Claimed in 4.2 *Goal Learning Strategy* that supervising for the group instead of the goal can eliminate instability brought from inaccurate value estimation. The “Impact of Group Selection” shows a curve with “no group selection”, how exactly was this done?
  2. Ablation for the value learning strategy: compare updating from transitions drawn uniformly from the replay buffer, without using $\mathcal{D}_{related}$. Along with the above, would allow you to show that the various components of GSRL are necessary
2. Can you clarify/explain more about the self-attention mechanism used in Eq 2? Is this similar to self-attention used in Transformer layers?

**After rebuttal responses**:

The authors have tried to address my concerns about the baselines and clarifications on some of the details of their implementation during the rebuttal period. After the revision, the additional ablations and GoalGAN/CHER baselines added to the empirical evidence for their algorithm. I updated my score to a weak acceptance.

**References**

[1]. Florensa et al.. Automatic goal generation for reinforcement learning agents. ICML, 2018

[2]. Hartikainen et al. Dynamical distance learning for semi-supervised and unsupervised skill discovery. In ICLR, 2020.

[3]. Pitis et al. Maximum Entropy Gain Exploration for Long Horizon Multi-goal Reinforcement Learning. ICML, 2020

[4]. Pong et al. Skew-Fit: State-Covering Self-Supervised Reinforcement Learning. ICML, 2020

[5]. Zhao et al.  Maximum entropy regularized multi-goal reinforcement learning. ICML, 2019

---

> ### Author Response · Authors · 2020-11-20
> **Response to Review #4**
>
> Thank you for your feedback and suggested experiments. Please also see the main comment above. We hope you will consider increasing your score after seeing our responses.
>
> >> There are several works that investigate letting the agent choose its own goals during exploration: GoalGAN [1], DDL [2], MEGA [3], Skew-Fit [4]. Rather than constructing a graph to determine the boundaries/frontier, they use either a discriminator or a density estimator. It would be worth discussing these works in relation to GSRL, and perhaps even including some of them as baselines.
>
> Thanks for your suggestion. We compare GSRL with GoalGAN (called GoalGAN in Figure 6) in the experiment section. We included all the previous works that you mention and discuss them in the related work section.
>
> >> For prioritizing experience during value estimation, there is also MEP which tries to group trajectories that achieve diverse goals during the update.
>
> Thanks for pointing it out. We discuss the relations and differences between GSRL and MEP in the related work section.
>
> >> How does the method perform without using the attention strategy on the groups to pick CATT first, then obtain the optimal goal within CATT?  The “Impact of Group Selection” shows a curve with “no group selection”, how exactly was this done? Claimed in 4.2 Goal Learning Strategy that supervising for the group instead of the goal can eliminate instability brought from inaccurate value estimation.
>
> Thanks for your question. In the experiment part, the baseline called NOGroup in Figure 6(g) (entitled “Impact of Group Selection”) is proposed to evaluate the performance of GSRL without using the attention strategy on the groups to pick C_ATT first, then obtain the optimal goal within C_ATT. In other words, NOGroup directly selects a visited state with the highest value as the goal. Since GSRL selects a group instead of a state, we can separate the learning of goal generation and value estimation (see Section 3.2). In other words, whether the value of the state is well estimated or not has no impact on the learning of goal generation (see Eqs. 2, 3, and 5). So, we claim that this learning strategy can eliminate instability brought from inaccurate value estimation and verify it in the experiment.
>
> >> Ablation for the value learning strategy: compare updating from transitions drawn uniformly from the replay buffer, without using Drelated. Along with the above, would allow you to show that the various components of GSRL are necessary.
>
> Thanks for your suggestion. We design a baseline (named ALL in Figure 6(i)) in the experiment section. This method draws transitions uniformly from the replay buffer, without using Drelated. Please see Figure 6(i) for details.
>
> >> Can you clarify/explain more about the self-attention mechanism used in Eq 2? Is this similar to self-attention used in Transformer layers?
>
> Thanks for your question. We have provided a detailed description of the self-attention mechanism in Section 3.1 Attention Strategy. It is similar to the one used in the Transformer layer. We show the formulation of the self-attention mechanism in Eq. 2. Please see Section 3.1 Attention Strategy for details.
>
> >> Some parts of the paper were confusing to read. In Section 3, the goal-oriented RL framework was introduced, with both discrete and continuous action space. I was confused then later in section 4 as the dynamic graph construction appears to rely on discrete actions in order to have a notion of degree of node s.
>
> Thanks for your question. We provide an illustrated example to discretize the continuous space into the discrete space in Appendix C. We also study the impact of using different discretization settings in Figure 6(h). Please see Appendix C for the illustration.

---

> > ### Comment · AnonReviewer4 · 2020-11-22
> > **Thank you for the response. Some additional questions**
> >
> >  Thank you to the authors for responding to my clarification questions and providing additional ablation results. I have some additional comments below:
> >
> > > We compare GSRL with GoalGAN (called GoalGAN in Figure 6) in the experiment section. We included all the previous works that you mention and discuss them in the related work section.
> >
> > Thank you for adding GoalGAN (and CHER) to the Ant Maze experiment, confirming the strengths of GSRL. For the other works discussed in the related work section, I am not entirely sure that the three additional works mentioned fits the current group of HRL works described as “discovering sub-tasks or sub-goals that are easy to reach in a short time and can guide the agent to the terminal state.” They are certainly un/semi-supervised objectives for discovering/choosing sub-goals to pursue though.
> >
> > > We discuss the relations and differences between GSRL and MEP in the related work section.
> >
> > Can briefly mention in what way GSRL is different from MEP (e.g. MEP maximizing entropy of selected trajectories).
> >
> > > In the experiment part, the baseline called NOGroup in Figure 6(g) (entitled “Impact of Group Selection”) is proposed to evaluate the performance of GSRL without using the attention strategy on the groups to pick C_ATT first, then obtain the optimal goal within C_ATT.
> >
> > It seems that NOGroup takes off much more slowly than using the C_ATT + optimal goal in C_ATT in the Fig 6.g. Asymptotically, it appears from the graph that NOGroup has the highest success rate? Or is that an artifact from the plotting when their values are all very similar/same. If asymptotically using NOGroup is better then perhaps having some sort of curriculum to switch/anneal from using the C_ATT + optimal goal in C_ATT could be interesting, but certainly outside of this rebuttal. One possible suggestion would be to have a separate numerical table of asymptotic performance values (with the uncertainty +/- values) in the Appendix in case some of these lines are too close to tell.
> >
> > > We design a baseline (named ALL in Figure 6(i)) in the experiment section. This method draws transitions uniformly from the replay buffer, without using Drelated. Please see Figure 6(i) for details.
> >
> > Thanks for adding this experiment. It seems that indeed $D_{related}$ is learning faster than uniform sampling from the replay buffer, although their asymptotic performance is very similar/same. For the final version, I would also recommend trying to have MEP as the baseline here as well, sampling from the replay buffer with high entropy in the trajectories.
> >
> > >  We have provided a detailed description of the self-attention mechanism in Section 3.1 Attention Strategy. It is similar to the one used in the Transformer layer. We show the formulation of the self-attention mechanism in Eq. 2. Please see Section 3.1 Attention Strategy for details.
> >
> > I have some additional questions about this, as it was not clear from the updated draft:
> > 1. To confirm, the features of groups $f_i$ is a $\mathbb{R}^k$ embedding vector that is learned independently for each of the $N$ groups?
> > 2. From my understanding of equation (2), $ATT_\phi(\mathcal{C}_1,\dots,\mathcal{C}_N)$ computes the new set of contextualized features of groups, say denoted as $[f’_1, \dots, f'_N]$, where $f’_i \in \mathbb{R}^{k’}$. Is $k’=1$ here? If not, how does this work for equation 3 then in order to select the group? I suspect you are trying to say that the sigmoid is applied to a $N$ dimensional vector which then allows you to take the argmax, but my understanding is that each $f’_i$ is not a scalar but a contextualized vector.
> >
> > > We provide an illustrated example to discretize the continuous space into the discrete space in Appendix C. We also study the impact of using different discretization settings in Figure 6(h). Please see Appendix C for the illustration.
> >
> > The figure in A.C made it more clear now about when the discretization happens (i.e. during the goal generation part, still continuous for replay buffer), thank you. It makes sense that small K bins is sufficient for the Maze environment, as probably it may even just require K=3 (top, middle, and bottom) for the U-shape.
> >
> > For the final revision, I would suggest to run the baselines on the Maze and FetchRun environments as well of course, but I understand for the short rebuttal phase there is only time to run the new baselines on one environment.

---

> > > ### Author Response · Authors · 2020-11-24
> > > **Response to Review #4**
> > >
> > > Thank you for your feedback and suggested experiments. We hope you will consider increasing your score after seeing our responses.
> > >
> > > > For the other works discussed in the related work section, I am not entirely sure that the three additional works mentioned fits the current group of HRL works described as “discovering sub-tasks or sub-goals that are easy to reach in a short time and can guide the agent to the terminal state.” They are certainly un/semi-supervised objectives for discovering/choosing sub-goals to pursue though.
> > >
> > > Thanks a lot for pointing it out. We have updated the related work section in the revision.
> > >
> > > > Can briefly mention in what way GSRL is different from MEP (e.g. MEP maximizing entropy of selected trajectories).
> > >
> > > We mention the difference between GSRL and MEP as follows: MEP maximizes entropy of selected trajectories; however, our method utilizes the structure information in the state-transition graph to select related trajectories for learning. We have updated the paper to further clarify this.
> > >
> > > > Asymptotically, it appears from the graph that NOGroup has the highest success rate? Or is that an artifact from the plotting when their values are all very similar/same. If asymptotically using NOGroup is better then perhaps having some sort of curriculum to switch/anneal from using the C_ATT + optimal goal in C_ATT could be interesting, but certainly outside of this rebuttal. One possible suggestion would be to have a separate numerical table of asymptotic performance values (with the uncertainty +/- values) in the Appendix in case some of these lines are too close to tell.
> > >
> > > Thanks for your suggestion. It is quite interesting to investigate designing some curriculum strategy to anneal from using the C_ATT + optimal goal in C_ATT.  The success rates of all the models (i.e., GSRL and its variants) are close to 1. Considering the potential fluctuation brought from the random seeds, it is difficult to tell which setting has the highest success rate. We provide a separate numerical table of Figure 6(g) in Appendix F.4. Figure 6(g) can imply that GSRL is more data efficient. We plan to run all the models in a more complex environment to make a deeper analysis of the attention module, and study which setting can obtain the highest success.
> > >
> > > > For the final version, I would also recommend trying to have MEP as the baseline here as well, sampling from the replay buffer with high entropy in the trajectories.
> > >
> > > Thanks for your suggestion. We have cited and discussed the relations and differences between MEP and GSRL. We only noticed MEP a few days ago. So, it is hard for us to finish the comparison with it. We will include this approach in the final version.
> > >
> > > > To confirm, the features of groups fi is a $\mathbb{R}^k$ embedding vector that is learned independently for each of the $N$ groups?
> > >
> > > Thanks for your question. We first construct the embedding vector for each group and update them through Eq.(5). Specifically, we concentrate all the embedding vectors of related groups into contextualized features and feed into the self-attention mechanism as formulated in Eq.(2). We have clarified this in the updated version.
> > >
> > > > I suspect you are trying to say that the sigmoid is applied to a $N$ dimensional vector which then allows you to take the argmax, but my understanding is that each $f'_i$ is not a scalar but a contextualized vector.
> > >
> > > Thanks for your question. It is correct that each fi is a d-dimensional embedding vector. Before the sigmoid, we adopt a MLP layer to project the output of the self-attention (i.e., a vector in $\mathbb{R}^{N \times d}$) into (i.e., a scalar in $\mathbb{R}^{N(\times 1)}$). We have corrected the attention module in the updated PDF, where the MLP layer is omitted in the previous version.  Thanks for pointing it out.
> > >
> > > > For the final revision, I would suggest to run the baselines on the Maze and FetchRun environments as well of course, but I understand for the short rebuttal phase there is only time to run the new baselines on one environment.
> > >
> > > Definitely, we will include all the environments in the final version. Unfortunately, it wasn’t possible to get results for all during short rebuttal time.

---

### Author Response · Authors · 2020-11-20
**Response to all reviewers**

We first summarize our response and the results of additional suggested experiments here. We have responded to the concerns of the reviewers as individual comments below.

All reviewers agree that our idea is interesting and our contributions are valuable to the community. To summarize our contributions: (i) we demonstrate that GSRL, a novel framework that leverage structure information of the state-transition graph for efficient goal generation and value estimation and  (ii) we adopt a self-attention mechanism in goal generation and design a novel learning algorithm to update it through a hindsight way.

Here is a list of new experiments:

1. Comparison with [1] (named CHER in Figure 6(b))

2. Comparison with [2] (named GoalGAN in Figure 6(b))

3. Ablation Study: GSRL without the group selection part [i.e., C_ATT] (named NOGroup in Figure 6(g)), GSRL without using related experience [i.e., D_related] (named ALL in Figure 6(i))

[1] Meng Fang, et al. Curriculum-guided Hindsight Experience Replay, NeurIPS, 2019.

[2] Florensa et al.. Automatic goal generation for reinforcement learning agents. ICML, 2018.

Note that we have updated Proposition 1 in the main text. In the original version, we cite [3] in the appendix. In the revision, we mention [3] in the main text since the main framework is borrowed from it. Since our task and detailed setting are considerably different, several modifications are required as described in Appendix D.1.

[3] Xiaoran Xu, et al. Dynamically pruned message passing networks for large-scale knowledge graph reasoning. ICLR, 2019.

---

### Decision · Program_Chairs · 2021-01-07
**Final Decision**

**Decision:**

Reject

**Comment:**

This work presents an algorithm - graph-structured reinforcement learning (GSRL)- to address the problem of exploration in sparse reward settings. The core elements of this work are 1) to build a state-transition graph from experienced trajectories in the replay buffer; 2) learn an attention module that chooses a goal from a subset of nodes in the graph and 3) policy learning via DDPG using "related trajectories", where trajectories that are related to the generated goal are sampled from the replay buffer.

Pros:
- all reviewers agree that the idea/work is interesting and valuable to the community
- reviewers appreciate the theoretical graph-based foundation/motivations

Cons:
- clarity: the manuscript still remains hard to follow. Many critical components for understanding are in the appendix.
-- One of the key steps in this work is the discretization of the state/action space for graph construction. However, this is not mentioned very clearly, which creates a lot of confusion given that you're considering continuous control domains.
-- Furthermore, the group selection part and training the attention module is expressed in an overly complex manner. Without the reviewers inquiries it would have been impossible to decode the technical details of this key contribution, and unfortunately it remains hard to read/follow.
-- while the ablation experiments (impact of discretization, group selection ..) are appreciated, but it is not clear on which environment they were generated (average across all? or only one of them?).
-- do you use DQN and DDPG? There are some conflicting statements in your paper, namely first you say "We use off-policy algorithms named DQN (Mnih et al., 2013) for discrete action space and DDPG (Lillicrap et al., 2015) for continuous action space", then in the experiments you say "to demonstrate the real performance gain of our GSRL we set the policy network with DDPG for GSRL and all baselines".
- I agree with the reviewers that it's not clear why the chosen baselines are very relevant - there seem to be other more relevant baselines.
- the significance of the attention module is not very clear, and is not analysed properly. What does it really learn? some form of deeper analysis would be useful here. How would a version that simply picks the most uncertain state in the graph? The ablation graph presented is not very convincing.


Overall, I believe that this work will make a valuable contribution in the future, with an iteration to improve clarity and better show-case the significance of the attention module.